# Vulnerable Agent Identification in Large-Scale Multi-Agent Reinforcement Learning

**Simin Li** [1 2] **Zihao Mao** [3] **Zheng Yuwei** [1] **Linhao Wang** [1] **Ruixiao Xu** [1] **Chengdong Ma** [4] **Zhiqian Liu** [1] **Xin Yu** [5]
**Yuqing Ma** [3] **Xin Wang** [3] **Jie Luo** [3] **Bo An** [6] **Yaodong Yang** [4] **Weifeng Lv** [1] **Xianglong Liu** [1]

## Abstract

Partial agent failure becomes inevitable when systems scale up, making it crucial to identify the subset of agents whose failure causes worst-case system performance degradations. We study this Vulnerable Agent Identification (VAI) problem in large-scale multi-agent reinforcement learning (MARL). We frame VAI as a Hierarchical Adversarial Decentralized Mean Field Control (HAD-MFC), where the upper level selects vulnerable agents as an NP-hard task and the lower level learns their worst-case adversarial policies via mean-field MARL. The two problems are coupled together, making HAD-MFC difficult to solve. To handle this, we first decouple the hierarchical process by Fenchel-Rockafellar transform, resulting a regularized mean-field Bellman operator for upper level that enables independent learning at each level, thus reducing computational complexity. We next reformulate the upper-level NP-hard problem as an MDP with dense rewards, allowing sequential identification of vulnerable agents via greedy and RL algorithms. This decomposition provably preserves the optimal solution. Experiments show our method effectively identifies more vulnerable agents in large-scale MARL and the rule-based system, fooling system into worse failures, and reveals the vulnerability of each agent in large systems. Code available at https://github.com/Waken-dream/VAI.

---

[1]School of Computer Science and Engineering, Beihang University [2]Department of Computer Science and Engineering, The Chinese University of Hong Kong [3]School of Artificial Intelligence, Beihang University [4]Institute for Artificial Intelligence, Peking University [5]Institute of Automation, Chinese Academy of Science [6]College of Computing and Data Science, Nanyang Technological University. Correspondence to: Xianglong Liu <xlliu@buaa.edu.cn>, Yaodong Yang <yaodong.yang@pku.edu.cn>.

*Proceedings of the 43rd International Conference on Machine Learning*, Seoul, South Korea. PMLR 306, 2026. Copyright 2026 by the author(s).

## 1. Introduction

Mean-field multi-agent reinforcement learning (MARL) (Yang et al., 2018; Subramanian et al., 2022; Pasztor et al., 2021; Laurière et al., 2022) has significantly enhanced the scalability of MARL through mean-field approximation, making it applicable to many large-scale real-world applications, such as robot swarm control (Hüttenrauch et al., 2019; Zheng et al., 2018), voltage control (Wang et al., 2021), and traffic control (Nguyen et al., 2018). However, given the large number of agents in such systems, it is likely that a small portion will deviate from the original policy during real-world deployment. For instance, in a thousand-robot swarm, individual robots may encounter action uncertainty (Tessler et al., 2019) from software or hardware errors (Khalastchi & Kalech, 2019), environmental hazards (Huang et al., 2019), or even be controlled by adversaries (Giray, 2013; Ly & Ly, 2021; Gleave et al., 2019; Lin et al., 2020; Dinh et al., 2023). These individual failures can ultimately lead to the failure of the entire team (Li et al., 2023a); In a power grid with hundreds of nodes (Wang et al., 2021), failure of certain nodes can trigger cascading failures, leading to a large-scale blackout (Liu et al., 2022). As agent policies are interconnected in mean-field MARL, it's crucial for defenders to evaluate the impact of the worst-case failure of a small group of agents on the entire system.

In this paper, we focus on vulnerable agent identification (VAI) in large-scale MARL systems. VAI is a proactive attack to identify the most vulnerable agents in a system. Given the set of most vulnerable agents, we further evaluate the system's worst-case robustness under adversarial attacks (Gleave et al., 2019), offering practitioners the worst-case performance of the system.

Critics may argue that vulnerable agents do not exist, as theoretical Mean-Field Controls assume all agents take identical actions (Lasry & Lions, 2007; Pasztor et al., 2021). However, in real-world large-scale MARL systems, agents often have different initializations, local states, or interact with limited neighbors (Zheng et al., 2018; Yang et al., 2018), leading to agent variability. In such cases, a mean-field approximation remains relevant but does not assume full agent homogeneity. Research in network science has

tackled *influence maximization* (Kempe et al., 2003; Banerjee et al., 2020; Li et al., 2023c), which seeks to select a group of nodes in rule-based social networks to maximize their influence. However, these studies typically assume known graph structures, transition dynamics, and influence rules, which are absent in our setting. Identifying vulnerable agents has also been explored in small-scale MARL systems (Pham et al., 2022; Zan et al., 2023; Zhou & Liu, 2023). The primary challenge arises from scale: a 10-agent system has only $\binom{10}{1}$ possible scenarios, while a 1000-agent system yields $\binom{1000}{100}$ scenarios, an increase by a factor of $10^{139}$. This represents a coupled problem where the upper level is a combinatorial problem, and the lower level involves mean-field MARL, making the complexity the central difficulty.

We begin by analyzing the complexity of the problem, which we formulate as a Hierarchical Adversarial Decentralized Mean Field Control (HAD-MFC). At the upper level, the task is to select $K$ most vulnerable agents from a total of $N$, resulting in a combinatorial problem with complexity $\binom{N}{K}$. We show that this problem is NP-hard by reducing it to the generalized maximum coverage problem (Cohen & Katzir, 2008). The lower level involves a mean-field MARL task, where an adversarial policy (Gleave et al., 2019) is trained for the selected $K$ vulnerable agents to assess the system's worst-case robustness. Consequently, the overall challenge requires solving an NP-hard upper-level problem and a coupled lower-level mean-field MARL task.

To solve the coupled HAD-MFC, we first decouple the upper-level attacker with the lower-level one. Given a selected set of vulnerable agents at the upper level, we propose regularized mean-field Bellman operator that estimate the potential worst-case value without running lowere-level attack. The operator is derived by applying the Fenchel-Rockafellar transform (Rockafellar, 1970) to lower-level uncertainties. To solve the NP-hard upper-level problem, we formulate it as a MDP with dense rewards calculated from value functions at upper-level, and solve it by greedy and RL algorithms. We prove this decomposition preserves the optima of HAD-MFC. Experiments on large-scale MARL and rule-based systems show that our method outperforms baselines in 17 out of 18 tasks, successfully identifying critical vulnerabilities and reveals the vulnerability of each agent in large-scale systems.

**Contributions.** Our contributions are twofold. First, we address the robustness of large-scale MARL by proposing the problem of vulnerable agent identification (VAI), formulating it as a HAD-MFC, and analyzing its hardness. Second, we show that HAD-MFC can be solved by decomposing the hierarchical process into two separate problems via Fenchel-Rockafellar transform and solve the upper-level NP-hard problem via formulating it as a MDP with dense reward.

## 2. Related Work

**Learning Large-Scale MARL**. In MARL, modeling the interactions between individual agents becomes impractical as the number of agents increases, making conventional MARL ineffective in large-scale (Yang & Wang, 2020). Mean-Field Games (MFGs) (Huang et al., 2006; Lasry & Lions, 2007) offer a solution by modeling the overall distribution of agents, instead of individual agents. Recent advances in equilibrium learning for MFGs (Guo et al., 2019; Perolat et al., 2021; Laurière et al., 2022; Muller et al., 2022; Carmona et al., 2023) have established strong theoretical foundations. Mean-Field Control (MFC) serves as the cooperative counterpart to MFGs (Gu et al., 2021; Mondal et al., 2022; Angiuli et al., 2022). Both frameworks assume a scenario where an infinite number of agents follow the same action distribution forming an mean field. However, in practical settings, agents need to take different actions based on their local states or specific policies. To address this, Yang et al. (2018) extended the mean-field approximation to Markov games by modeling opponents through an action mean field using a Taylor expansion. This approach has been expanded to accommodate various MARL settings, including stationary (Subramanian & Mahajan, 2019), multi-type (Subramanian et al., 2020a), and partially observable environments (Subramanian et al., 2020b). A more structured framework, known as decentralized MFGs (Subramanian et al., 2022), has also been developed, with significant contributions from Sessa et al. (2022); Cui et al. (2023; 2024). Our study utilizes this decentralized framework, which has been proven to be highly effective in large-scale MARL (Zheng et al., 2018).

**Adversarial Attacks for MARL**. The goal of adversarial attacks for MARL is to develop worst-case adversarial attacks of MARL under uncertainties. This includes uncertainties in state (Lin et al., 2020; Zan et al., 2023; Zhou & Liu, 2023), action (Guo et al., 2022; Li et al., 2023a), or environment (Zhang et al., 2020; Shi et al., 2024) to cause a well-trained MARL algorithm to fail during testing. Among these studies, several focus on selecting the most vulnerable agents to attack. For instance, GMA-FGSM (Zan et al., 2023) groups agents by their features and selects vulnerable agents based on their contribution to the total reward. ARTS (Phan et al., 2020) evaluates system robustness by repeatedly selecting random groups of agents to act as attackers. The work most similar to ours is RTCA (Zhou & Liu, 2023), which employs a differential evolution algorithm to select vulnerable agents. However, these approaches are confined to small-scale MARL, and the challenge of scaling them to large-scale MARL remains unexplored.

**Influence Maximization**. First proposed by Kempe et al. (2003), influence maximization involves selecting a set of nodes in a social network to influence the opinions of others

through predefined rules. Kempe et al. (2003) demonstrated that this problem is NP-hard and introduced a greedy algorithm to solve it. Early works relied on heuristics, such as degree centrality (Chen et al., 2009; Wilson et al., 2009), graph structure (Chen et al., 2010; Cordasco et al., 2015), genetic algorithms (Tsai et al., 2015; Bucur & Iacca, 2016), and community-based methods (Wang et al., 2010; Chen et al., 2014). More recent works address the problem by combining graph neural networks and reinforcement learning, learning a network embedding that serves as input to an RL algorithm for sequential node selection (Meirom et al., 2021; Li et al., 2022; Chen et al., 2023). In contrast to these approaches, Ling et al. (2023) demonstrated the potential to learn directly from network embeddings. However, most influence maximization studies assume a *known* graph, transition dynamics, and operate within a rule-based system. Our work does not rely on any of these assumptions.

## 3. Problem Formulation

### 3.1. Hierarchical Adversarial Decentralized Mean-Field Control

We formulate our problem as a Hierarchical Adversarial Mean-Field Control (HAD-MFC). To model large-scale MARL that assumes heterogeneous agents with mean-field approximations, we base our definition on decentralized Mean-Field Control (D-MFG) (Subramanian et al., 2022). HAD-MFC adapts D-MFG by fixing the victim policy and training an adversarial policy to (1) select a subset of agents from the victim agents, (*i.e.* agents not being attacked) and (2) replace the selected agents' policies with a worst-case adversarial policy. The HAD-MFC is defined as follows:

$$\mathcal{G} := \langle \mathcal{N}, \mathcal{S}, \mathcal{A}, \mathcal{P}, R, \mu_0, \nu_0, \gamma \rangle, \tag{1}$$

where $\mathcal{N} = \{1, \ldots, N\}$ represents the set of $N$ agents, $\mathcal{S}$ and $\mathcal{A}$ denote the finite state and action spaces for each agent. $\mathcal{P} : \mathcal{S} \times \mathcal{A} \times \Delta(\mathcal{S}) \times \Delta(\mathcal{A}) \to \Delta(\mathcal{S})$ is the state transition probability function, $R : \mathcal{S} \times \mathcal{A} \times \Delta(\mathcal{S}) \times \Delta(\mathcal{A}) \to \mathbb{R}$ is the shared reward function, $\mu_0 \in \Delta(\mathcal{S})$ and $\nu_0 \in \Delta(\mathcal{A})$ are the initial state and action distributions, and $\gamma \in [0, 1)$ is the discount factor. The interactions between agents are modeled through the mean-field state $\Delta(\mathcal{S})$ and action distribution $\Delta(\mathcal{A})$ in both the environment dynamics and rewards.

Let $\mathcal{T} = \{0, 1, \ldots, T\}$ represent the set of time steps. At $t = 0$, the attacker selects $k$ agents to form an attack set $\mathcal{K}$, where $\mathcal{K} \subseteq \mathcal{N}$ and $|\mathcal{K}| = k$, which remains fixed in the episode. At each time step $t \in \mathcal{T}$, each agent $i$ receives a local state $s_t^i \in \mathcal{S}$ and estimates the empirical mean-field state $\mu_t(s) = \frac{1}{N} \sum_{j \in \mathcal{N}} \delta(s_t^j = s)$, with $\delta$ the Dirac's delta. Each agent first executes a fixed, well-trained cooperative policy $\pi_\beta(a_t^i | s_t^i, \mu_t) : \mathcal{S} \times \Delta(\mathcal{S}) \to \Delta(\mathcal{A})$. To model the policy deviation under uncertainty, we assign a perturbation budget $\epsilon^i \in [0, 1]$ for each agent. If agent $i$ is in attack set

$\mathcal{K}$, the adversary learns an adversarial action perturbation policy $\pi_\alpha(a_t^i | s_t^i, \mu_t) : \mathcal{S} \times \Delta(\mathcal{S}) \to \Delta(\mathcal{A})$, and yields a perturbed policy $\hat{\pi}^i = \epsilon^i \pi_\alpha^i + (1 - \epsilon^i) \pi_\beta^i \in \Delta(\mathcal{A})$, following the definition of PR-MDP in (Tessler et al., 2019). Here, $\epsilon^i$ limits the deviation of agents from the original policy, while assuming that attackers do not have access to the victim's policy. If agent $i$ is fully controlled by the attacker, this corresponds to the case where $\epsilon^i = 1$. If agent $i$ is not in attack set $\mathcal{K}$, the victim executes $\hat{\pi} = \pi_\beta$ with $\epsilon^i = 0$. The empirical mean-field action is $\nu_t(a) = \frac{1}{N} \sum_{j \in \mathcal{N}} \delta(a_t^j = a)$. The reward at time $t$ is given by $r_t = R(\{s_t^i\}_{i \in \mathcal{N}}, \{a_t^i\}_{i \in \mathcal{N}}, \mu_t, \nu_t) \equiv R(s_t, a_t, \mu_t, \nu_t)$, which is shared across agents. The game then transitions to time $t + 1$, generating a new local state for each agent based on the environment transition $p(s_{t+1}^i | s_t^i, a_t^i, \mu_t, \nu_t)$. The expected reward is:

$$J(\hat{\pi}) \equiv J(\pi_\alpha, \pi_\beta) = \mathbb{E}_{\pi_\alpha, \pi_\beta} \left[ \sum_{t=0}^{\infty} \gamma^t R(s_t, a_t, \mu_t, \nu_t) \right]. \tag{2}$$

**Attacker's goal.** The attacker's goal is to select an attack set $\mathcal{K}$ such that the agents in $\mathcal{K}$ learn an adversarial policy to minimize the expected reward:

$$\min_{\mathcal{K} \subseteq \mathcal{N}, |\mathcal{K}| = k} \min_{\pi_\alpha} J(\pi_\alpha, \pi_\beta). \tag{3}$$

**Complexity issue.** The attacker face a hierarchical problem. The upper level face a combinatorial problem to select the $k$ most vulnerable agents, and the lower level learns an adversarial policy for these selected agents. The coupled nature characterize the complexity issue of our problem.

**Relation to existing formulations.** Our definition of HAD-MFC is distinct yet related to several existing formulations in the literature. Our study focus on control of practical large-scale MARL with mean-field approximation (Subramanian et al., 2022; Mondal et al., 2022) rather than theoretical MFGs and MFCs (Guo et al., 2019; Muller et al., 2022; Gu et al., 2021), and specifically focuses on the selection of vulnerable agents rather than equilibrium learning and optimal agent control. Our upper-level problem of selecting vulnerable agents is conceptually similar to influence maximization (IM) (Kempe et al., 2003). However, unlike IM, where influencing agents follow predefined rules, our framework requires agents to learn an adversarial policy and to cooperate optimally with other adversarial agents. Our lower-level problem is related to adversarial attacks in MARL (Gleave et al., 2019). Existing works either do not involve the selection of vulnerable agents (Lin et al., 2020; Li et al., 2023a), or are limited to small-scale settings (Pham et al., 2022; Zhou & Liu, 2023). Our approach addresses adversarial attacks in large-scale MARL environments using mean-field approximations, which are significantly more complex than previously studied methods.

## 3.2. Assumptions and Theoretical Analysis

In this section, we outline the assumptions underlying our attack model. Building on existing studies on adversarial MARL (Tessler et al., 2019; Gleave et al., 2019; Li et al., 2023b; Dinh et al., 2023), we introduce a practical threat model based on specific assumptions regarding the capabilities of both victims and attackers at different levels.

**Assumption 3.1** (Victim's capability). *Victims follow a fixed, well-trained policy $\pi_\beta$ that remains unchanged during the attack.*

We assume that the victim policies are fixed to simulate an attack scenario at test time, where the large-scale MARL system is deployed and its policy does not adapt in response to the attack (Tessler et al., 2019; Gleave et al., 2019). Thus, victims compute their empirical mean field over all agents, including attackers. The attacker's goal is to minimize the victims' reward given that victims continue to act based on this mean-field signal.

**Assumption 3.2** (Upper-level attacker's capabilities and limitations). *The upper-level attacker can select $k$ agents from $\mathcal{N}$ and assign individual perturbation budgets $\epsilon^i, i \in \mathcal{K}$ only at the beginning of an episode. The upper-level attacker has access to* all *agents' trajectories under the* cooperative *case, $\tau = [\{s_0^i\}_{i \in \mathcal{N}}, \{a_0^i\}_{i \in \mathcal{N}}, \mu_0, \nu_0, r_0, \ldots, \{s_T^i\}_{i \in \mathcal{N}}, \{a_T^i\}_{i \in \mathcal{N}}, \mu_T, \nu_T, r_T]$. During the attack, it can also access the local state $\{s_t^i\}_{i \in \mathcal{N}}$ of all agents at $t = 0$ and the cumulative reward $r = \sum_{t \in \mathcal{T}} \gamma^t r_t$. It does not have access to the policy parameters of the victim agents.*

**Proposition 3.3** (Hardness). *The problem faced by the upper-level attacker is NP-hard.*

*Proof sketch.* We prove this by reducing the maximum coverage problem, which is known to be NP-hard, to our upper-level attack. See full proof in Appendix A.1.

**Assumption 3.4** (Lower-level attacker's capabilities and limitations). *The lower-level attacker $\min_{\pi_\alpha} J(\pi_\alpha, \pi_\beta)$ has access to its local state $s_t^i$, the empirical mean field $\mu_t, \nu_t$, and the reward $r_t$. It does not have access to the policies, value functions, or local states of other agents.*

Our upper-level attacker only requires access to cooperative trajectory data, which is relatively easy to obtain. Furthermore, our attack model is *black-box* for both upper-level and lower-level, without the need of victim's policy (note that for lower-level attacker, its policy is added on, yet irrelevant to victim policy). Lastly, we establish the existence of an optimal adversary.

**Proposition 3.5** (Existence of optimal adversary). *For any HAD-MFC, there exists an optimal (i.e., most harmful) upper-level adversary $\mathcal{K}$ and a corresponding lower-level adversary $\pi_\alpha$.*

*Proof sketch.* The upper-level attack is a finite combinatorial problem with an optimal solution. At the lower level, with fixed victim policies treated as part of the environment, the attacker solves a MFC problem with optimal solution. The optimal adversary exists by exploring all upper-level configurations and selecting the best lower-level policy. See full proof in Appendix. A.2. $\square$

# 4. Method

In this section, we propose algorithms to solve the complexity issue of HAD-MFC. We begin by decoupling the hierarchical problem, eliminating the need to train a worst-case lower-level adversary by reformulating it into a regularized mean-field Bellman operator. We then formulate the upper-level combinatorial task as a MDP with dense reward computed from the value function from the regularized mean-field Bellman operator, and solve it via greedy algorithm or RL.

## 4.1. Decoupling the Hierarchical Problem

Training the worst-case adversary $\pi_\alpha$ is computationally expensive since it requires solving the RL problem $\min_{\pi_\alpha} J(\pi_\alpha, \pi_\beta)$. To address this, we propose a regularized mean-field Bellman operator that efficiently estimates the value function under a worst-case adversary, using cooperative trajectories only. Our approach involves defining the Bellman function for the adversary, characterizing the uncertainty set induced by $\pi_\alpha$, and applying Fenchel-Rockafellar transform to derive the solution.

**Bellman operators.** To begin, we define the value function $V^i(s^i, \mu)$ for our problem:

$$V^i(s^i, \mu) = \mathbb{E}\left[\sum_{t=0}^{\infty} \gamma^t r_t \middle| s_0 = s, \mu_0 = \mu, a_t^i \sim \hat{\pi}(\cdot|s_t^i, \mu_t)\right].$$
(4)

The Bellman operator $\mathcal{B}^{\hat{\pi}}$ with victim and adversary policy can be defined as:

$$(\mathcal{B}^{\hat{\pi}} V^i)(s^i, \mu) = \sum_{a \in \mathcal{A}} \hat{\pi}(a^i|s^i, \mu)\nu(a)\Big[r +$$
$$\gamma \sum_{s' \in \mathcal{S}} p(s'^i|s^i, a^i, \mu, \nu)V(s'^i, \mu')\Big].$$
(5)

With worst-case adversary, we can further define the worst-case Bellman operator as:

$$(\hat{\mathcal{B}}^{\hat{\pi}} V^i)(s^i, \mu) = \min_{\pi_\alpha}(\mathcal{B}^{\hat{\pi}} V^i)(s^i, \mu)$$
(6)

**Uncertainty set characterization.** We proceed by characterizing the impact of $\pi_\alpha$ on perturbed policy $\hat{\pi}$ and the perturbed mean-field action $\nu(a)$. We expand them as:

$$\hat{\pi}^i = \epsilon^i \pi_\alpha^i + (1 - \epsilon^i)\pi_\beta^i,$$
$$\lim_{N \to \infty} \nu(a) = \xi \nu_\alpha(a) + (1 - \xi)\nu_\beta(a),$$
(7)

where $\quad \xi = \frac{1}{N}\sum_{i\in\mathcal{N}}\epsilon^i, \nu_\alpha(a) = \frac{1}{N}\sum_{i\in\mathcal{N}}\delta(a_t^i = a|\pi_\alpha),$

$$\nu_\beta(a) = \frac{1}{N}\sum_{i\in\mathcal{N}}\delta(a_t^i = a|\pi_\beta).$$

We can then derive the uncertainty set induced by $\pi_\alpha$:

**Proposition 4.1.** The difference of perturbed policy and victim policy, perturbed mean-field action and victim mean-field action can be (approximately) bounded in $\ell_p$ norm:

$$\forall \delta > 0 : ||\hat{\pi}^i - \pi_\beta^i||_p \leq 2^{1/p}\epsilon^i,$$

$$p\left(\left|||\nu(a) - \nu_\beta(a)||_p - 2^{1/p}\xi\right| \geq \delta\right) \leq 2\exp\left(-2N\delta^2/2^{2/p}\right)$$

*Proof sketch.* The proof for $\hat{\pi}$ is by expanding itself and $||\pi_\alpha - \pi_\beta||_p \leq 2^{1/p}$. The proof for $\nu$ is by Jensen's inequality and the probability is by Hoeffding's inequality. Since the factor $2^{1/p}$ is a constant independent of the parameters, we absorb it into $\epsilon^i$ and $\xi$ in subsequent derivations to avoid cluttered expression, without loss of generality. See full proof in Appendix.A.3.

**Fenchel-Rockafellar transform.** With uncertainty set defined, we simplify the notation by $\hat{\pi}_\alpha^i = \hat{\pi}^i - \pi_\beta^i$ and $\hat{\nu}_\alpha(a) = \nu(a) - \nu_\beta(a)$, which is bounded by $||\hat{\pi}_\alpha^i||_p \leq \epsilon^i$ and $\hat{\nu}_\alpha(a) \lesssim \xi$ by Proposition. 4.1. We proceed by expanding the Bellman equation in Eqn. 6:

$$(\mathcal{B}^{\hat{\pi}}V^i)(s^i,\mu) = \sum_{a^i,a\in\mathcal{A}}\left(\hat{\pi}_\alpha^i + \pi_\beta^i\right)\left(\hat{\nu}_\alpha(a) + \nu_\beta(a)\right)\Big[r_t$$

$$+ \gamma\sum_{s'\in\mathcal{S}}p(s'^i|s^i,a^i,\mu,\nu)V(s'^i,\mu')\Big].$$

Proposition 3.5 ensures optimal adversary always exists. With $(\hat{\mathcal{B}}^{\hat{\pi}}V^i)(s^i,\mu) = \min_{\pi_\alpha}(\mathcal{B}^{\hat{\pi}}V^i)(s^i,\mu)$, we then have the following robust Bellman inequality (Iyengar, 2005):

$$V^i(s^i,\mu) = (\hat{\mathcal{B}}^{\hat{\pi}}V^i)(s^i,\mu) \leq (\mathcal{B}^{\hat{\pi}}V^i)(s^i,\mu),$$
$$V^i(s^i,\mu) - (\mathcal{B}^{\hat{\pi}}V^i)(s^i,\mu) \leq 0, \qquad (8)$$

with equality holds when $\pi_\alpha$ reach optimality $\pi_\alpha^*$. Thus, we are solving the following problem via Fenchel-Rockafellar transform (Rockafellar, 1970; Nachum & Dai, 2020):

$$\max_{\pi_\alpha} V^i(s^i,\mu) - (\mathcal{B}^{\hat{\pi}}V^i)(s^i,\mu). \qquad (9)$$

**Proposition 4.2.** The Fenchel-Rockafellar transform of Eqn. 9 results in:

$$\max_{\pi_\alpha} V^i(s^i,\mu) - (\mathcal{B}^{\hat{\pi}}V^i)(s^i,\mu) = V^i(s^i,\mu)$$
$$- \mathcal{B}_{\epsilon^i,\xi}^R V^i(s^i,\mu,\epsilon^i,\xi) = V^i(s^i,\mu) - (\mathcal{B}^{\pi_\beta}V^i)(s^i,\mu)$$
$$+ (\epsilon^i + \xi + \epsilon^i\xi)||Q^i(s^i,a^i,\mu,\nu)||_q.$$

A change of variable yields the regularized mean-field Bellman operator $\mathcal{B}_{\epsilon^i,\xi}^R$:

$$\mathcal{B}_{\epsilon^i,\xi}^R V^i(s^i,\mu,\epsilon^i,\xi) = (\mathcal{B}^{\pi_\beta}V^i)(s^i,\mu)$$
$$- (\epsilon^i + \xi + \epsilon^i\xi)||Q^i(s^i,a^i,\mu,\nu)||_q. \qquad (10)$$

Here, $1/p + 1/q = 1$ is the dual of $\ell_p$ norm via Fenchel-Rockafellar transform. In this way, our learned value function $V^i(s^i,\mu,\epsilon^i,\xi)$ estimated from our Bellman estimator $\mathcal{B}_{\epsilon^i,\xi}^R$ quantifies agent $i$'s performance under attack, condition on two factors: (1) the agent's own perturbation status $\epsilon^i$, and (2) the mean-field approximation on $\xi$, which indicates the number of its teammates gets perturbed.

*Proof sketch.* We first expand $\hat{\pi}$ and $\nu(a)$ in Eqn. 6, resulting in a Q function with uncertainty. Applying Fenchel-Rockafellar transform completes the proof. See full proof in Appendix. A.4. $\square$

**Proposition 4.3** (Contraction). The regularized mean-field Bellman operator $\mathcal{B}_{\epsilon^i,\xi}^R V^i(s^i,\mu,\epsilon^i,\xi) = (\mathcal{B}^{\pi_\beta}V^i)(s^i,\mu) - (\epsilon^i + \xi + \epsilon^i\xi)||Q^i(s^i,a^i,\mu,\nu)||_q$ is a contraction operator.

*Proof sketch.* To proof that, we find $||Q^i(s^i,a^i,\mu,\nu)||_q$ term cancels each other and the rest follows standard approach. See full proof in Appendix. A.5. $\square$

**Proposition 4.4** (Relation to worst-case Q function). As a geometric intuition, we show $\epsilon^i\xi||Q^i(s^i,a^i,\mu,\nu)||_q$ corresponds to the first-order deviation between the cooperative and worst-case Q-functions under $\ell_p$-bounded perturbed action $a_\alpha^i$ and mean-field action $\nu_\alpha$ induced by $\pi_\alpha$:

$$\epsilon^i\xi||Q^i(s^i,a^i,\mu,\nu)||_q = \max_{||a_\alpha^i||_p \leq \epsilon^i, ||\nu_\alpha||_p \leq \xi}$$
$$||Q^i(s^i,a^i,\mu,\nu) - Q^i(s^i,(a^i + a_\alpha^i),\mu,(\nu + \nu_\alpha))||_1.$$

*Proof sketch.* The proof is done by making a linear approximation of Q function, then applying Hölder's inequality. See full proof in Appendix. A.6. *Note that this approximation serves solely as a post-hoc explanation to visualize the regularizer's effect. The validity of our main result in Proposition 4.2 is guaranteed to be exact by the Fenchel-Rockafellar transform and does not rely on this approximation.*

**Remark 1.** The regularization terms in $\mathcal{B}^R$ arises from uncertainties in agents and the mean-field. To clarify, the term $\epsilon^i||Q^i(s^i,a^i,\mu,\nu)||_q$ capture agent vulnerability, $\xi||Q^i(s^i,a^i,\mu,\nu)||_q$ capture mean-field vulnerability, and $\epsilon^i\xi||Q^i(s^i,a^i,\mu,\nu)||_q$ capture vulnerability of their interactions. Each term yields more pessimistic value estimation when there are larger uncertainties in its actions, mean-field, or their interactions.

**Remark 2.** Notably, our approach does not assume $\pi_\beta$ to be optimal, which means it can be extended to agent-based systems governed by predefined rules (An et al., 2021), provided these rules can be derived from Q-functions (e.g., using a Boltzmann-based policy).

**Remark 3.** *(Exactness and approximation sources.)* Two distinct value functions appear in our framework, and it is important to distinguish their roles and approximation statuses clearly. The first is the *robust V function*

$V^i(s^i, \mu, \epsilon^i, \xi)$, learned via the regularized mean-field Bellman operator $\mathcal{B}^R_{\epsilon^i,\xi}$ and used as the upper-level reward proxy (Eqn. 12). The second is the *cooperative Q function* $Q^i(s^i, a^i, \mu, \nu)$, which is learned solely under the cooperative victim policy $\pi_\beta$, without any adversarial perturbation. This Q function enters $\mathcal{B}^R_{\epsilon^i,\xi}$ as a fixed dual term arising from the Fenchel–Rockafellar transform; it is pre-computed from cooperative trajectory rollouts $\tau \sim \pi_\beta$ and held fixed during the learning of $V^i$. Crucially, the Fenchel–Rockafellar transform in Proposition 4.2 is *exact*: it requires only that the uncertainty set be convex, proper, and lower semicontinuous, all of which are satisfied by our $\ell_p$-bounded rectangular set, as established in Proposition 4.1. The transform itself introduces no approximation error; nor does it require the value function or policy to be convex. All approximation errors in practice therefore have two and only two sources: (1) function approximation error in learning $Q^i$ and $V^i$ with neural networks, and (2) the sub-optimality of the NP-hard upper-level combinatorial search.

### 4.2. Algorithm for Vulnerable Agent Identification

To solve the NP-hard upper-level problem of identifying vulnerable agents, we formulate the task as a MDP with dense rewards from our regularized mean-field Bellman operator. We approximate the problem via greedy algorithms and RL. Finally, we show our decomposition is lossless, preserving the optima of HAD-MFC.

**Problem formulation.** The problem faced by the upper-level adversary can be formulated as a Markov Decision Process, defined based on HAD-MFC:

$$\mathcal{M} := \langle \boldsymbol{S}, \epsilon, \mathcal{N}, \tilde{\mathcal{P}}, \tilde{R}, \gamma \rangle, \tag{11}$$

where $\boldsymbol{S} = \times_{i \in \mathcal{N}} \mathcal{S}^i$ is the local state space of each agent. The game proceeds in $K$ steps, with $K$ the number of adversaries we select. At step $k$, $\epsilon_k \in [0,1]^N = \{\epsilon^i_k\}_{i \in \mathcal{N}}$ is the perturbation budget of each agent, with $\epsilon^i_0 = 0, \forall i \in \mathcal{N}$. $\mathcal{N}$ is the action space, where agents could be selected as vulnerable agent, $\tilde{\mathcal{P}} : \boldsymbol{S} \times \mathcal{N} \to \boldsymbol{S}$ is the state transition, and $\tilde{R} : \boldsymbol{S} \times \mathcal{N} \times [0,1] \to \mathbb{R}$ is the reward function, $\gamma$ is the discount factor. At each step $k$, we select the most vulnerable agent $n$, and update the value of $\epsilon_k$. Note that the problem becomes a standard MDP if we merge $\epsilon$ in $\mathcal{S}$.

**Reward.** Reward specifies the objective of MDP. In our case, the reward is defined as: given the set of selected vulnerable agents $\mathcal{K}_{k-1}$ and the new selected agent $n_k$ at step $k$, what is the amount of reward the victim large-scale MARL system going to decrease, had it face the worst-case adversary trained on this new set of selected vulnerable agents $\mathcal{K}_k = \mathcal{K}_{k-1} \cup n_k$?

To calculate this value efficiently, we resort to the regularized mean-field Bellman operator $\mathcal{B}^R_{\epsilon^i,\xi}$ in Eqn.10, which defines the amount of reward we expected to receive, given

the $\ell_p$ bounded perturbation magnitude $\epsilon^i_k$ and $\xi_k$ at step $k$. Define the value function learned under $\mathcal{B}^R_{\epsilon^i,\xi}$ at time $t = 0$ as $V^i(s^i_0, \mu_0, \epsilon^i_k, \xi_k)$, the reward can then be defined as:

$$r_k = \tilde{R}(s_k, \epsilon_k, n_k) = \frac{1}{N} \sum_{i \in \mathcal{N}} \big( \tag{12}$$
$$V^i(s^i_0, \mu_0, \epsilon^i_{k-1}, \xi_{k-1}) - V^i(s^i_0, \mu_0, \epsilon^i_k, \xi_k) \big).$$

Here $\epsilon^i_k$ can take any values between $[0, 2^{1/p}]$ and $\xi_k$ depends on $\epsilon^i_k$. We thus define the TD loss as:

$$\min_{\mathbb{E}_{\tau \sim \pi_\beta}} (V^i(s^i, \mu, \epsilon^i, \xi) - r - \gamma V^i(s'^i, \mu', \epsilon^i, \xi)$$
$$+ (\epsilon^i \xi + \epsilon^i + \xi)||Q^i(s^i, a^i_\beta, \mu, \nu_\beta)||_q)^2, \tag{13}$$

with $\epsilon \sim Uniform[0, 2^{1/p}]$, $\xi \sim Bernoulli(\xi)$. The value function can be optimized by collected trajectory rollouts in cooperative case using victim policy (i.e., $\tau \sim \pi_\beta$), which can be easy to obtain.

**Solving the MDP.** Given the RL formulation, we can optimize our VAI problem using any RL algorithm, such as DQN (Mnih et al., 2015a), and updates the Q function via standard TD loss. We call this approach as VAI-RL. Alternatively, the reward defined in Eqn. 12 suggests a fast greedy algorithm, which selects the agent to maximize reward at each step. We call this approach VAI-Greedy. We include both algorithms for comparison, with pseudo code in Appendix. B.

**Proposition 4.5** (Decomposition is Optimality-Preserving). Given a HAD-MFC $\mathcal{G} := \langle \mathcal{N}, \mathcal{S}, \mathcal{A}, \mathcal{P}, R, \mu_0, \nu_0, \gamma \rangle$. For the upper-level MDP $\mathcal{M} := \langle \boldsymbol{S}, \epsilon, \mathcal{N}, \tilde{\mathcal{P}}, \tilde{R}, \gamma \rangle$ with reward defined in Eqn.12, and the value $V^{i,*}(s^i, \mu, \epsilon^i, \xi)$ of lower-level problem is learned by regularized mean-field Bellman operator $\mathcal{B}^R_{\epsilon^i,\xi}$, define the optimal vulnerable agents of $\mathcal{M}$ as $\mathcal{K}^* \subseteq \mathcal{N}$. The selected vulnerable agents $\mathcal{K}^* \subseteq \mathcal{N}$ and the worst-case adversarial policy learned $\pi^*_\alpha$ under $\mathcal{K}^* \subseteq \mathcal{N}$ is the optimal solution of HAD-MFC.

*Proof sketch.* We prove this by showing the optimal solution of lower- and upper-level is the same as original HAD-MFC. The lower-level transformation is lossless because the Fenchel–Rockafellar transform with rectangular uncertainty set. The upper-level problem is a MDP with optimal solution. See full proof in Appendix. A.7.

## 5. Experiments

**Environments.** We evaluate our algorithms in three environments. Battle (Zheng et al., 2018) and Taxi Matching (Nguyen et al., 2018) are mean-field MARL tasks with discrete control, and Vicsek (Vicsek et al., 1995) is a rule-based system with continuous control. Detailed descriptions of the ewnvironments are provided in Appendix C.1. We train all victim agents in Battle using MF-Q, and Taxi Matching

*Table 1.* VAI consistently achieves superior attack performance across three environments, with varying map sizes and attacker numbers.

| AGENT NUM | ADV. NUM | RANDOM | DC | BI-LEVEL | PIANO | RTCA | VAI-GREEDY | VAI-RL |
|---|---|---|---|---|---|---|---|---|
| | | | | ENVIRONMENT: BATTLE ($\downarrow$) | | | | |
| 64 | 8 | $298.47_{\pm76.56}$ | $305.16_{\pm45.39}$ | $295.09_{\pm12.96}$ | $296.79_{\pm47.67}$ | $301.08_{\pm22.72}$ | $\mathbf{287.53_{\pm9.39}}$ | $\mathbf{281.50_{\pm17.33}}$ |
| | 16 | $97.33_{\pm34.52}$ | $93.54_{\pm34.56}$ | $87.37_{\pm6.28}$ | $81.06_{\pm11.34}$ | $85.71_{\pm24.62}$ | $\mathbf{72.01_{\pm20.28}}$ | $\mathbf{77.73_{\pm1.81}}$ |
| | 32 | $-152.89_{\pm26.75}$ | $-160.51_{\pm75.32}$ | $-198.03_{\pm55.83}$ | $-175.24_{\pm39.11}$ | $-192.78_{\pm43.81}$ | $-214.40_{\pm43.12}$ | $\mathbf{-929.88_{\pm62.73}}$ |
| 144 | 18 | $730.65_{\pm117.42}$ | $693.15_{\pm98.87}$ | $685.77_{\pm124.51}$ | $670.55_{\pm66.75}$ | $650.33_{\pm50.47}$ | $\mathbf{610.62_{\pm31.36}}$ | $\mathbf{505.34_{\pm30.79}}$ |
| | 36 | $250.43_{\pm120.19}$ | $140.67_{\pm76.67}$ | $189.95_{\pm15.54}$ | $130.63_{\pm34.69}$ | $155.02_{\pm170.74}$ | $\mathbf{85.52_{\pm35.11}}$ | $86.26_{\pm38.72}$ |
| | 72 | $-1809.01_{\pm130.98}$ | $-2014.57_{\pm670.92}$ | $-2353.78_{\pm870.53}$ | $-2313.46_{\pm230.66}$ | $-2221.12_{\pm360.49}$ | $-2579.80_{\pm256.19}$ | $\mathbf{-2837.83_{\pm482.56}}$ |
| | | | | ENVIRONMENT: TAXI ($\downarrow$) | | | | |
| 50 | 4 | $33.9_{\pm14.39}$ | $19.07_{\pm5.77}$ | $27.52_{\pm16.12}$ | $23.55_{\pm7.44}$ | $16.26_{\pm3.32}$ | $\mathbf{10.47_{\pm4.85}}$ | $\mathbf{12.47_{\pm8.73}}$ |
| | 16 | $109.94_{\pm7.32}$ | $79.01_{\pm11.33}$ | $162.23_{\pm2.31}$ | $140.60_{\pm49.01}$ | $138.73_{\pm1.72}$ | $\mathbf{54.63_{\pm8.81}}$ | $64.72_{\pm3.76}$ |
| | 36 | $617.09_{\pm51.80}$ | $595.80_{\pm60.28}$ | $571.26_{\pm59.96}$ | $516.91_{\pm44.86}$ | $618.21_{\pm54.08}$ | $463.70_{\pm55.99}$ | $\mathbf{365.96_{\pm63.75}}$ |
| 100 | 4 | $34.49_{\pm22.61}$ | $21.17_{\pm3.47}$ | $14.17_{\pm3.07}$ | $36.51_{\pm6.11}$ | $16.87_{\pm8.27}$ | $\mathbf{8.27_{\pm8.67}}$ | $\mathbf{4.95_{\pm2.86}}$ |
| | 16 | $172.00_{\pm75.41}$ | $141.19_{\pm5.80}$ | $201.14_{\pm68.66}$ | $202.51_{\pm47.18}$ | $\mathbf{140.76_{\pm32.44}}$ | $153.97_{\pm8.52}$ | $186.62_{\pm40.79}$ |
| | 36 | $884.49_{\pm68.87}$ | $867.62_{\pm23.46}$ | $892.51_{\pm66.15}$ | $793.71_{\pm12.86}$ | $860.58_{\pm106.61}$ | $770.14_{\pm29.74}$ | $\mathbf{652.10_{\pm23.23}}$ |
| | | | | ENVIRONMENT: VICSEK ($\uparrow$) | | | | |
| 100 | 20 | $-226.96_{\pm11.54}$ | $-232.45_{\pm3.77}$ | $-221.26_{\pm14.06}$ | $-250.83_{\pm19.59}$ | $-225.12_{\pm28.05}$ | $\mathbf{-167.60_{\pm3.91}}$ | $-183.68_{\pm19.56}$ |
| | 35 | $-159.83_{\pm40.85}$ | $-143.14_{\pm42.37}$ | $-141.51_{\pm43.28}$ | $-162.74_{\pm28.45}$ | $-129.24_{\pm13.30}$ | $\mathbf{-113.64_{\pm15.78}}$ | $-93.65_{\pm28.65}$ |
| | 50 | $-96.83_{\pm7.26}$ | $-95.22_{\pm6.19}$ | $-96.80_{\pm0.76}$ | $-86.21_{\pm3.55}$ | $-82.63_{\pm5.70}$ | $\mathbf{-70.52_{\pm5.21}}$ | $-75.82_{\pm2.57}$ |
| 400 | 80 | $-884.34_{\pm53.96}$ | $-840.87_{\pm33.67}$ | $-780.31_{\pm90.02}$ | $-950.13_{\pm110.36}$ | $-872.21_{\pm130.11}$ | $\mathbf{-710.56_{\pm56.32}}$ | $-659.65_{\pm86.73}$ |
| | 140 | $-480.17_{\pm50.16}$ | $-440.63_{\pm80.67}$ | $-460.43_{\pm74.71}$ | $-510.24_{\pm62.11}$ | $-410.14_{\pm87.33}$ | $\mathbf{-390.74_{\pm42.16}}$ | $-302.76_{\pm76.37}$ |
| | 200 | $-295.13_{\pm36.94}$ | $-313.55_{\pm49.43}$ | $-310.78_{\pm56.89}$ | $-290.53_{\pm27.89}$ | $-287.53_{\pm46.76}$ | $\mathbf{-256.44_{\pm21.34}}$ | $-275.62_{\pm37.76}$ |

using MF-AC (Yang et al., 2018), which empirically yields better task performance.

**Baselines.** To our knowledge, the problem of vulnerable agent identification in MARL is rarely studied in literature. Therefore, we select five relevant studies as baselines: (1) Random selection, serving as a simple baseline. (2) Degree centrality (DC) (Salathé & Jones, 2010), a heuristic method that select agents with the most connections with others. (3) Bi-level RL (Vezhnevets et al., 2017), which trains our upper-level and lower-level problems hierarchically. (4) PIANO (Li et al., 2022), which selects critical agents iteratively via graph embeddings and RL. (5) RTCA (Zhou & Liu, 2023), which selects vulnerable agents in small-scale MARL using differential evolution. For methods requiring a graph structure, we construct an undirected graph with an edge of weight 1 between two agents if they can observe each other, and 0 otherwise. We call our method as Vulnerable Agent Identification (VAI). All baselines are trained using the same codebase, network structure, and hyperparameters for fair comparison. Detailed implementations and hyperparameters are provided in Appendix C.2 and C.3.

**Evaluation protocol.** We consider $\{\epsilon^i\}_{i\in\mathcal{K}} = 1$ bounded by $\ell_\infty$ norm, allowing adversaries to manipulate the policy of $\pi_\beta$ arbitrarily. The scenario occurs when agents crash in the environment, or are compromised by the adversary (Khalastchi & Kalech, 2019; Huang et al., 2019; Gleave et al., 2019). Results of different $\epsilon$ in Appendix. D.2. The number of attackers, $K$, is empirically determined based on the total number of agents in the environment. We report the results on victims and attackers with five random seeds.

### 5.1. Simulation Results

First, we evaluate the effectiveness of our method on finding the most vulnerable agents to attack. This is done by (1) solve the upper-level problem of finding the most vulnerable agents and (2) solve the lower-level problem of learning a worst-case policy for these vulnerable agents. For compre-

hensiveness, for each task, we evaluate them on six subtasks, including two map sizes with different numbers of agents and adversaries.

As shown in Table 1, our VAI based method **outperforms all baselines in 17 out of 18 tasks**, while heuristic-based method and learning based method are only slightly better than random selection. To explain this, heuristic-based method such as degree centrality (DC) are designed for rule-based systems. However, in large-scale MARL, the interaction between agents are nonlinear and are not clearly defined by rules. For example, in Battle environment, agents in the center of the crowd are less susceptible to attack than the agents in frontline, combating enemies, making DC ineffective. As for learning-based method, PIANO do not account for the worst-case policy made by agents, thus are unable to select the set of most harmful agents under adversarial policies. Solving our problem via bi-level RL and RTCA do not work well due to the hierarchical nature of our problem, which may be too hard for RL to solve without any guidance. In contrast, our VAI method works well due to the more accurate value function we learned via Bellman operator $\mathcal{B}^R_{\epsilon^i,\xi}$.

Additionally, we observed that VAI-RL outperforms VAI-Greedy in 10 of 18 tasks, especially when more attackers are available. To explain, greedy algorithm focuses on immediate reward and works well with less attackers and weak agent-wise interactions. RL, in contrast, models long-term returns and inter-agent impact, which performs better with more attackers. Additionally, RL provides theoretical guarantees for optimality in MDPs, which greedy methods lack.

Finally, our VAI algorithm yields superior results on **both MARL and rule-based systems**. In rule-based environments, we approximate a value function from collected trajectories, then use our Bellman operator to estimate each agent's vulnerability. Our work could have a future impact on rule-based complex system with real-world impact, such as social networks(Banerjee et al., 2020).

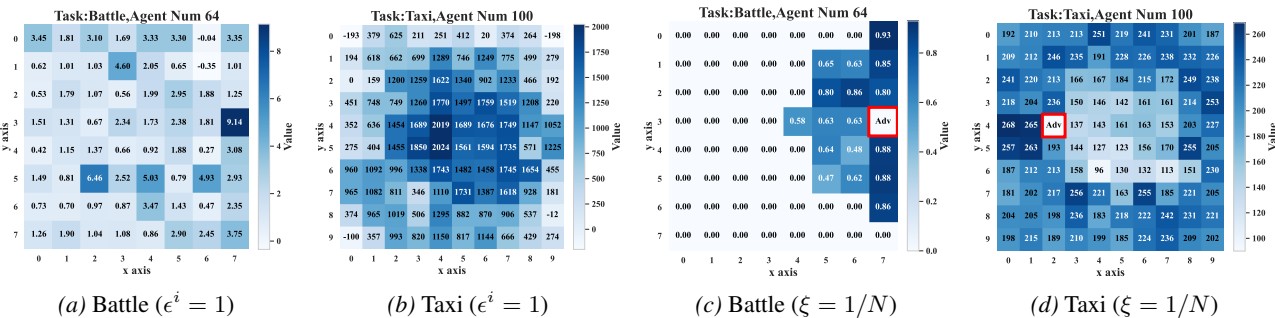

*(a)* Battle ($\epsilon^i = 1$)      *(b)* Taxi ($\epsilon^i = 1$)      *(c)* Battle ($\xi = 1/N$)      *(d)* Taxi ($\xi = 1/N$)

*Figure 1.* (a,b) Change from $\epsilon^i = 0$ to $\epsilon^i = 1$. Some agents contributes more to overall system when compromised. (c,d) Change from $\xi = 0$ to $\xi = 1/N$. Agents receive more impact when facing attackers. Darker cell indicates higher vulnerability.

**Computational Efficiency:** While our VAI requires training an additional function in Proposition 4.2, we find the overall computation cost is on par with baselines. See full results in Appendix. D.1.

**Results with Different $\epsilon$:** Under different perturbation budget $\epsilon$, our VAI-RL and VAI-Greedy consistently outperform other baselines. See full results in Appendix. D.2.

### 5.2. Discussions and Insights

In this section, we thoroughly evaluate the effectiveness of our method, showing our theory is effective in practice and our method offers meaningful insights to the robustness of large-scale MARL.

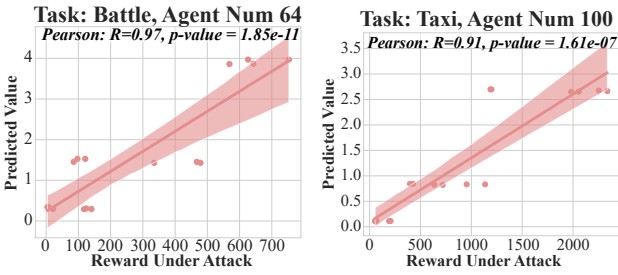

*Figure 2.* Pearson correlation between the lower-level attack value estimated by our Bellman operator $\mathcal{B}^R_{\epsilon^i, \xi}$ (y axis) and lower-level reward by running an attack using RL (x axis). Our proposed Bellman operator is an accurate predictor of attack result.

**Our Method is Effective by Proposed Value Estimation in Proposition. 4.2.** Our regularized mean-field Bellman operator $\mathcal{B}^R_{\epsilon^i, \xi}$ is the key to our success. To verify this, we compare the results predicted by our value function of the lower-level attack, and the reward gained by actually running the lower-level attack via RL. As shown in Fig. 2, we find the value function learned by $\mathcal{B}^R_{\epsilon^i, \xi}$ is effective at predicting the attack result of the worst-case adversarial policy for vulnerable agent selections, showing strong Pearson correlation ($r = 0.97$ for Battle, $r = 0.91$ for Taxi, $p < .001$). Thus, $\mathcal{B}^R_{\epsilon^i, \xi}$ effectively decomposes HAD-MFC by acting as a predictor of lower-level attack

**Our Method Reveals Certain agents are more vulnerable than others.** In large-scale MARL systems, some agents

play disproportionately critical roles, making them inherently more vulnerable. To illustrate this, we visualize agent values in the Battle-64 and Taxi-100 environments using heatmaps in Fig. 1, where each cell represents the importance of a single agent at the start of the game. In Battle-64, 64 agents are arranged in an 8×8 grid to engage with another team of 64 agents; we display only the 64 agents controlled by the mean-field MARL policy. In Taxi-100, 100 agents are uniformly positioned across a 10×10 map. The heatmaps reveal two key factors on vulnerability:

**First, some agents contribute more significantly to overall system functionality.** Figs. 1 (a,b) visualize the value difference $V^i(s^i, \mu, \epsilon^i = 0, \xi = 0) - V^i(s^i, \mu, \epsilon^i = 1, \xi = 0)$, reflecting the drop in value if agent $i$ is selected as an adversary, as captured by the $\epsilon^i$ term in Proposition 4.2. In Battle, agents at the right hand side engage enemies more frequently and thus accumulate more rewards, making them both more valuable and more vulnerable when compromised. In Taxi, ride requests occur more frequently near the center, so agents located there earn higher rewards and are more critical. These patterns indicate that agents with advantageous positions or key roles contribute more to cooperation and are therefore prime targets for adversarial exploitation.

**Second, the failure of one agent can negatively affect others.** Figs. 1 (c,d) show the impact of a single adversary (highlighted in a red square) on its teammates' value functions, computed as $V^i(s^i, \mu, \epsilon^i = 0, \xi = 0) - V^i(s^i, \mu, \epsilon^i = 0, \xi = 1/N)$, corresponding to the $\xi$ term in Proposition 4.2. In Battle, disruption primarily affects agents in the same row. An adversarial agent can mislead allies moving towards different directions, and disrupting the collective attacks to lower success rate. In Taxi, agents to the left suffer most. These agents must move toward the central region with the highest reward, but are actively blocked by the adversary, preventing them from reaching these high-reward areas. These results demonstrate that the learned $V^i$ function captures inter-agent dependencies and accurately reflects vulnerability propagation within the team.

# 6. Conclusions

In this paper, we identify vulnerable agents in large-scale MARL. We formulate the problem as a HAD-MFC, where the upper level attack addresses the NP-hard problem of selecting the most vulnerable agents, while the lower level attack learns worst-case adversarial policies. We disentangle this hierarchical problem using Fenchel-Rockafellar transform and solve the NP-hard upper-level problem with greedy algorithm and RL. Experiments show that our method identifies groups of vulnerable agents in both large-scale MARL and rule-based systems, causing these systems to experience worst-case failures when attacking these agents. Our method also learns a value function that accurately predicts the vulnerability of each agent. Future work will extend VAI to complex real-world domains, including social networks and economic agent-based models.

# Impact Statement

Our research focuses on the critical security problem of identifying vulnerable agents in large-scale MARL. The primary positive impact of this work is to provide system developers and administrators with a diagnostic tool identify the weakest points of a system. While attackers could potentially use our method to attack the weakest spot of the system, our method requires access to system trajectories, which can be hard for attackers to obtain, but easier for defenders. Our theoretical framework also suggests future work on robust large-scale MARL. We thus believe the benefit of our work outweighs potential security threats.

# Acknowledgments

This work is supported by National Natural Science Foundation of China (No.62525601) and Young Elite Scientists Sponsorship Program of the Beijing High Innovation Plan (No.20250627).

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

# APPENDIX FOR "VULNERABLE AGENT IDENTIFICATION IN LARGE-SCALE MULTI-AGENT REINFORCEMENT LEARNING"

## A. Proofs and Derivations

### A.1. Proof of Proposition 3.3

To prove the NP-hardness of our upper-level attack, we show the problem can be reduced from the maximum coverage problem, which is known to be NP-hard.

**Maximum Coverage Problem.** A maximum coverage problem is defined by a universe of elements $\mathcal{U} = \{e_1, \ldots, e_n\}$, a collection of subsets $\mathcal{S} = \{S_1, \ldots, S_m\}$ where $S_i \subseteq \mathcal{U}$, and an integer $k$. The objective is to select a sub-collection $\mathcal{S}' \subseteq \mathcal{S}$ with $|\mathcal{S}'| \leq k$ that maximize the number of covered elements, i.e., $\max_{\mathcal{S}' \subseteq \mathcal{S}, |\mathcal{S}'|=k} |\bigcup_{S_i \in \mathcal{S}'} S_i|$.

**Reduction from Maximum Coverage Problem.** We construct a mapping from the maximum coverage problem to our upper-level attack as follows. Let the set of agents $\mathcal{N}$ correspond one-to-one with the collection of subsets $\mathcal{S}$, such that selecting agent $i$ as a vulnerable agent corresponds to selecting the subset $S_i$. Selecting $k$ vulnerable agents corresponds to selecting $k$ subsets in the maximum coverage problem. We construct a MARL environment where the system reward without attack is defined as the total weight of all elements, $R_{total} = |\mathcal{U}|$. If an agent $i$ is attacked (i.e., $i \in \mathcal{K}$), it disables all elements in $S_i$, the attacker's reward is then $r = R_{total} - |\bigcup_{i \in \mathcal{K}} S_i|$. Our attacker's objective is to minimize the reward $r$ over choices of attacked agents $\mathcal{K} \subseteq \mathcal{N}$ with $|\mathcal{K}| \leq k$. Since $R_{total}$ is a constant, minimizing $r = R_{total} - |\bigcup_{i \in \mathcal{K}} S_i|$ is equal to maximizing $|\bigcup_{i \in \mathcal{K}} S_i|$. Thus, choosing an optimal attack $\mathcal{K}^*$ in our upper-level attack is exactly equivalent to choosing an optimal subset $\mathcal{S}'$ in the maximum coverage problem. Since the maximum coverage problem is NP-hard, our upper-level attack is NP-hard. $\qquad\square$

### A.2. Proof of Proposition 3.5

To prove the existence of an optimal adversary for our hierarchical problem, we demonstrate that optimal solutions exist for both the upper-level and lower-level attackers. The optimal adversary strategy can be determined by enumerating all possible configurations for the upper-level attacker and find the optimal policy for the lower-level attacker.

**Step 1: Lower-Level Attacker**

For the lower-level attacker, the identities of the victim and the adversary agents are fixed. The policy of the victim agents, denoted by $\prod_{i \in \mathcal{N}} \pi_\beta(a^i|s^i, \mu)$, is also fixed. This allows us to incorporate the victim's policy into the environment's transition dynamics, resulting in a modified transition function given by:

$$p'(s'^i|s^i, a^i, \mu, \nu) = p(s'^i|s^i, a^i, \mu, \nu) \cdot \prod_{i \in \mathcal{N}} \pi_\beta(a^i|s^i, \mu).$$

The lower-level attacker thus faces a new MFC problem with these modified environment dynamics $p'$. According to the results established in Carmona et al. (2018), an optimal policy exists for MFC problems, ensuring that the lower-level attacker can achieve an optimal strategy.

**Step 2: Upper-Level Attacker**

The upper-level attacker faces a finite combinatorial problem, as it involves selecting $k$ agents from a total of $N$ agents, leading to $\binom{N}{k}$ possible combinations. The optimal solution can be determined by exhaustively enumerating all possible combinations of agents and evaluating the corresponding outcomes.

For each combination of agents selected by the upper-level attacker, we train an MFC policy for the lower-level attacker. Given that an optimal policy is guaranteed for the lower-level problem, each combination of upper-level selections results in a cumulative reward.

Since the upper-level problem is a finite combinatorial optimization problem, there exists an optimal solution that maximizes the cumulative reward. Thus, the optimal adversary strategy consists of the optimal set of agents selected by the upper-level attacker, combined with the optimal lower-level policy determined by the MFC problem. Therefore, an optimal adversary exists for this hierarchical problem. $\qquad\square$

## A.3. Proof of proposition. 4.1

**(1) Bound for $\hat{\pi}$.**

As $\hat{\pi}^i = \epsilon^i \pi_\alpha^i + (1 - \epsilon^i)\pi_\beta^i$, we can expand $||\hat{\pi}^i - \pi_\beta^i||_p$ by:

$$||\hat{\pi}^i - \pi_\beta^i||_p = ||\epsilon^i \pi_\alpha^i + (1 - \epsilon^i)\pi_\beta^i - \pi_\beta^i||_p = ||\epsilon^i(\pi_\alpha^i - \pi_\beta^i)||_p = \epsilon^i||(\pi_\alpha^i - \pi_\beta^i)||_p \leq 2^{1/p}\epsilon^i. \tag{14}$$

Note that $||(\pi_\alpha^i - \pi_\beta^i)||_p \leq 2^{1/p}$. $\qquad\square$

**(2) Bound for $\nu$.**

We first consider the case when $N \to \infty$. In this case,

$$\lim_{N\to\infty} ||\nu(a) - \nu_\beta(a)||_p = ||\frac{1}{N}\sum_{i\in\mathcal{N}}\left(\hat{\pi}^i - \hat{\pi}_\beta^i\right)||_p = \frac{1}{N}||\sum_{i\in\mathcal{N}}\left(\hat{\pi}^i - \hat{\pi}_\beta^i\right)||_p. \tag{15}$$

Applying Jensen's inequality, we have:

$$\frac{1}{N}||\sum_{i\in\mathcal{N}}\left(\hat{\pi}^i - \hat{\pi}_\beta^i\right)||_p \leq \frac{1}{N}\sum_{i\in\mathcal{N}}||\left(\hat{\pi}^i - \hat{\pi}_\beta^i\right)||_p \leq \frac{1}{N}\sum_{i\in\mathcal{N}}2^{1/p}\epsilon^i = 2^{1/p}\xi. \tag{16}$$

Next, for finite $N$,

$$p\left(\left|||\nu(a) - \nu_\beta(a)||_p - \mathbb{E}\left[||\nu(a) - \nu_\beta(a)||_p\right]\right| \geq \delta\right) \tag{17}$$

$$=p\left(\left|\frac{1}{N}||\sum_{i\in\mathcal{N}}\delta(a^i = a|\hat{\pi}^i) - \delta(a^i = a|\pi_\beta^i)||_p - 2^{1/p}\xi\right| \geq \delta\right) \quad \text{(By Eqn. 16 )} \tag{18}$$

$$\leq p\left(\left|\frac{1}{N}\sum_{i\in\mathcal{N}}||\delta(a^i = a|\hat{\pi}^i) - \delta(a^i = a|\pi_\beta^i)||_p - 2^{1/p}\xi\right| \geq \delta\right) \quad \text{(By Jensen's inequality).} \tag{19}$$

Since $\delta(a^i = a|\hat{\pi})$ and $\delta(a^i = a|\pi_\beta)) \in \{0, 1\}$, we have each independent variable $\frac{1}{N}||\delta(a^i = a|\hat{\pi}^i) - \delta(a^i = a|\pi_\beta^i)||_p \leq \frac{2^{1/p}}{N}$. Thus, consider the largest deviation possible, by Hoeffding's inequality, $\forall \delta > 0$,

$$p\left(\left|\frac{1}{N}\sum_{i\in\mathcal{N}}||\delta(a^i = a|\hat{\pi}^i) - \delta(a^i = a|\pi_\beta^i)||_p - 2^{1/p}\xi\right| \geq \delta\right) \tag{20}$$

$$\leq 2\exp\left(-\frac{2\delta^2}{\sum_{i\in\mathcal{N}}(2^{1/p}/N)^2}\right) = 2\exp\left(-2N\delta^2/2^{2/p}\right) \tag{21}$$

To sum up, we have

$$p\left(\left|||\nu(a) - \nu_\beta(a)||_p - 2^{1/p}\xi\right| \geq \delta\right) \leq 2\exp\left(-2N\delta^2/2^{2/p}\right), \quad \forall \delta > 0. \tag{22}$$

This completes the proof. $\qquad\square$

## A.4. Proof of Proposition. 4.2

**Remark on the Fixed Q-function Assumption**: Consistent with our practical implementation (Algorithm 1), the following derivation focuses on the robust policy improvement step. We assume access to a fixed Q-value estimate $Q(s, a, \mu, \nu)$ obtained from the policy evaluation phase. Under this "fixed Q-function" assumption, the $\left[r_t + \gamma \sum_{s'\in\mathcal{S}} p(s'^i|s^i, a^i, \mu, \nu)V^i(s'^i, \mu')\right]$ term is treated as a constant coefficient vector with respect to the perturbations. This decouples the implicit dependence of transition dynamics on $\hat{\nu}$ for the purpose of the update rule derivation.

We begin our proof from Eqn. 9, using $\hat{\pi}_\alpha^i = \hat{\pi}^i - \pi_\beta^i$ as a shorthand, such that the perturbation budget is bounded by $\hat{\pi}_\alpha^i$ directly:

$$\max_{\pi_\alpha \in \mathcal{A}} V^i(s^i, \mu) - (\mathcal{B}^{\hat{\pi}} V^i)(s^i, \mu),$$

$$= \max_{\pi_\alpha \in \mathcal{A}} V^i(s^i, \mu) - \sum_{a^i, a \in \mathcal{A}} \left(\hat{\pi}_\alpha^i + \pi_\beta^i\right) \left(\hat{\nu}_\alpha(a) + \nu_\beta(a)\right) \left[ r_t + \gamma \sum_{s' \in \mathcal{S}} p(s'^i | s^i, a^i, \mu, \nu) V^i(s'^i, \mu') \right]$$

$$= \max_{||\hat{\pi}_\alpha^i||_p \leq \epsilon^i, ||\hat{\nu}_\alpha||_p \leq \xi} V^i(s^i, \mu) - \sum_{a^i, a \in \mathcal{A}} \left(\hat{\pi}_\alpha^i + \pi_\beta^i\right) \left(\hat{\nu}_\alpha(a) + \nu_\beta(a)\right) \Big[ r_t$$

$$+ \gamma \sum_{s' \in \mathcal{S}} p(s'^i | s^i a^i, \mu, \nu) V^i(s'^i, \mu') \Big]$$

$$= \max_{||\hat{\pi}_\alpha^i||_p \leq \epsilon^i, ||\hat{\nu}_\alpha||_p \leq \xi} V^i(s^i, \mu) - \sum_{a^i, a \in \mathcal{A}} \left(\hat{\pi}_\alpha^i \hat{\nu}_\alpha(a) + \hat{\pi}_\alpha^i \nu_\beta(a) + \pi_\beta^i \hat{\nu}_\alpha(a) + \pi_\beta^i \nu_\beta(a)\right) \Big[ r_t$$

$$+ \gamma \sum_{s' \in \mathcal{S}} p(s'^i | s^i, a^i, \mu, \nu) V^i(s'^i, \mu') \Big]$$

$$= V^i(s^i, \mu) - \min_{||\hat{\pi}_\alpha^i||_p \leq \epsilon^i, ||\hat{\nu}_\alpha||_p \leq \xi} \sum_{a^i, a \in \mathcal{A}} \hat{\pi}_\alpha^i \hat{\nu}_\alpha(a) \left[ r + \gamma \sum_{s' \in \mathcal{S}} p(s'^i | s^i, a^i, \mu, \nu) V^i(s'^i, \mu') \right]$$

$$- \min_{||\hat{\pi}_\alpha^i||_p \leq \epsilon^i} \sum_{a^i, a \in \mathcal{A}} \hat{\pi}_\alpha^i \nu_\beta(a) \left[ r + \gamma \sum_{s' \in \mathcal{S}} p(s'^i | s^i, a^i, \mu, \nu) V^i(s'^i, \mu') \right]$$

$$- \min_{||\hat{\nu}_\alpha||_p \leq \xi} \sum_{a^i, a \in \mathcal{A}} \hat{\pi}_\beta^i \nu_\alpha(a) \left[ r + \gamma \sum_{s' \in \mathcal{S}} p(s'^i | s^i, a^i, \mu, \nu) V^i(s'^i, \mu') \right]$$

$$- \sum_{a^i, a \in \mathcal{A}} \hat{\pi}_\beta^i \nu_\beta(a) \left[ r + \gamma \sum_{s' \in \mathcal{S}} p(s'^i | s^i, a^i, \mu, \nu) V^i(s'^i, \mu') \right].$$

Since the equation is too long, we analyze each separately. For the first line, we have:

$$\min_{||\hat{\pi}_\alpha^i||_p \leq \epsilon^i, ||\hat{\nu}_\alpha||_p \leq \xi} \sum_{a^i, a \in \mathcal{A}} \hat{\pi}_\alpha^i \hat{\nu}_\alpha(a) \left[ r + \gamma \sum_{s' \in \mathcal{S}} p(s'^i | s^i, a^i, \mu, \nu) V^i(s'^i, \mu') \right]$$

$$= \min_{\hat{\pi}_\alpha^i \in \mathcal{A}, \hat{\nu}_\alpha \in \mathcal{A}} \sum_{a^i, a \in \mathcal{A}} \hat{\pi}_\alpha^i \hat{\nu}_\alpha(a) \left[ r + \gamma \sum_{s' \in \mathcal{S}} p(s'^i | s^i, a^i, \mu, \nu) V^i(s'^i, \mu') \right]$$

$$= \min_{\hat{\pi}_\alpha^i \in \mathcal{A}, \hat{\nu}_\alpha \in \mathcal{A}} \sum_{a^i, a \in \mathcal{A}} \left(\hat{\pi}_\alpha^i \hat{\nu}_\alpha(a) + \delta_{||\hat{\pi}_\alpha^i||_p \leq \epsilon^i, ||\hat{\nu}_\alpha||_p \leq \xi}\right) \left[ r + \gamma \sum_{s' \in \mathcal{S}} p(s'^i | s^i, a^i, \mu, \nu) V^i(s'^i, \mu') \right].$$

$$= - \max_{\hat{\pi}_\alpha^i \in \mathcal{A}, \hat{\nu}_\alpha \in \mathcal{A}} \sum_{a^i, a \in \mathcal{A}} \left(\hat{\pi}_\alpha^i \hat{\nu}_\alpha(a) + \delta_{||\hat{\pi}_\alpha^i||_p \leq \epsilon^i, ||\hat{\nu}_\alpha||_p \leq \xi}\right) \left[ r + \gamma \sum_{s' \in \mathcal{S}} p(s'^i | s^i, a^i, \mu, \nu) V^i(s'^i, \mu') \right].$$

We can write it in the form needed by Fenchel-Rockafellar transform:

$$- \max_{\hat{\pi}_\alpha^i \in \mathcal{A}, \hat{\nu}_\alpha \in \mathcal{A}} \sum_{a^i, a \in \mathcal{A}} \left(\hat{\pi}_\alpha^i \hat{\nu}_\alpha(a) + \delta_{||\hat{\pi}_\alpha^i||_p \leq \epsilon^i, ||\hat{\nu}_\alpha||_p \leq \xi}\right) \left[ r + \gamma \sum_{s' \in \mathcal{S}} p(s'^i | s^i, a^i, \mu, \nu) V^i(s'^i, \mu') \right]$$

$$= - \max_{\hat{\pi}_\alpha^i \in \mathcal{A}, \hat{\nu}_\alpha \in \mathcal{A}} \langle \hat{\pi}_\alpha^i \hat{\nu}_\alpha, r + \gamma \sum_{s' \in \mathcal{S}} p(s'^i | s^i, a^i, \mu, \nu) V^i(s'^i, \mu') \rangle$$

$$+ \langle \delta_{||\hat{\pi}_\alpha^i||_p \leq \epsilon^i, ||\hat{\nu}_\alpha||_p \leq \xi}, r + \gamma \sum_{s' \in \mathcal{S}} p(s'^i | s^i, a^i, \mu, \nu) V^i(s'^i, \mu') \rangle$$

We can then apply Fenchel-Rockafellar transform:

$$f^*(-y) = \min_{\hat{\pi}_\alpha^i \in \mathcal{A}, \hat{\nu}_\alpha \in \mathcal{A}} -\langle \hat{\pi}_\alpha^i \hat{\nu}_\alpha, y \rangle - \langle \hat{\pi}_\alpha^i \hat{\nu}_\alpha, r + \gamma \sum_{s' \in \mathcal{S}} p(s'^i|s^i, a^i, \mu, \nu) V^i(s'^i, \mu') \rangle.$$

To maximize the following objective, we have:

$$y = -(r^i + \gamma \sum_{s' \in \mathcal{S}} p(s'^i|s^i, a^i, \mu, \nu) V^i(s'^i, \mu')) = -Q^i(s^i, a^i, \mu, \nu).$$

Plugging in $y$, we get:

$$- \max_{\hat{\pi}_\alpha^i \in \mathcal{A}, \hat{\nu}_\alpha \in \mathcal{A}} \langle \hat{\pi}_\alpha^i \hat{\nu}_\alpha, r + \gamma \sum_{s' \in \mathcal{S}} p(s'^i|s^i, a^i, \mu, \nu_\beta) V^i(s'^i, \mu') \rangle$$

$$+ \langle \delta_{||\hat{\pi}_\alpha^i||_p \leq \epsilon^i, ||\hat{\nu}_\alpha||_p \leq \xi}, r + \gamma \sum_{s' \in \mathcal{S}} p(s'^i|s^i, a^i, \mu, \nu) V^i(s'^i, \mu') \rangle$$

$$= \langle \delta_{||\hat{\pi}_\alpha^i||_p \leq \epsilon^i, ||\hat{\nu}_\alpha||_p \leq \xi}, Q^i(s^i, a^i, \mu, \nu) \rangle$$

$$= \epsilon^i \xi ||Q^i(s^i, a^i, \mu, \nu)||_q.$$

Similar to this derivation, other equations in our expanded form can be written as:

$$\min_{||\hat{\pi}_\alpha^i||_p \leq \epsilon^i} \sum_{a^i, a \in \mathcal{A}} \hat{\pi}_\alpha^i \nu_\beta(a) \left[ r + \gamma \sum_{s' \in \mathcal{S}} p(s'^i|s^i, a^i, \mu, \nu) V^i(s'^i, \mu') \right]$$

$$= \epsilon^i ||Q^i(s^i, a^i, \mu, \nu)||_q.$$

and

$$\min_{||\nu_\alpha||_p \leq \xi} \sum_{a^i, a \in \mathcal{A}} \hat{\pi}_\beta^i \nu_\alpha(a) \left[ r + \gamma \sum_{s' \in \mathcal{S}} p(s'^i|s^i, a^i, \mu, \nu) V^i(s'^i, \mu') \right]$$

$$= \xi ||Q^i(s^i, a^i, \mu, \nu)||_q$$

Note that the derivation processes are mostly the same, so we do not waste space on writing these very similar derivations.

Summing all these together, we get:

$$\max_{\pi_\alpha \in \mathcal{A}} V^i(s^i, \mu) - (\mathcal{B}^{\hat{\pi}} V^i)(s^i, \mu)$$

$$= V^i(s^i, \mu) - \sum_{a^i, a \in \mathcal{A}} \hat{\pi}_\beta^i \nu_\beta(a) \left[ r + \gamma \sum_{s' \in \mathcal{S}} p(s'^i|s^i, a^i, \mu, \nu) V^i(s'^i, \mu') \right] + (\epsilon^i + \xi + \epsilon^i \xi) ||Q^i(s^i, a^i, \mu, \nu)||_q$$

$$= V^i(s^i, \mu) - (\mathcal{B}^{\pi_\beta} V^i)(s^i, \mu) + (\epsilon^i + \xi + \epsilon^i \xi) ||Q^i(s^i, a^i, \mu, \nu)||_q.$$

This completes the proof. □

## A.5. Proof of Proposition. 4.3

Given the Bellman equation $\mathcal{B}_{\epsilon^i, \xi}^R V^i(s^i, \mu, \epsilon^i, \xi) = (\mathcal{B}^{\pi_\beta} V^i)(s^i, \mu) + (\epsilon^i + \xi + \epsilon^i \xi) ||Q^i(s^i, a^i, \mu, \nu)||_q$, let $V_1^i, V_2^i \in \mathbb{R}^{|\mathcal{S} \times \mathcal{S} \times [0,1] \times [0,1]|}$. Consider any $s \in \mathcal{S}, \mu \in \mathcal{S}, \epsilon^i \in [0,1], \xi \in [0,1]$, we have:

$$|\mathcal{B}_{\epsilon^i, \xi}^R V_1^i(s^i, \mu, \epsilon^i, \xi) - \mathcal{B}_{\epsilon^i, \xi}^R V_2^i(s^i, \mu, \epsilon^i, \xi)|$$

$$= |(\mathcal{B}^{\pi_\beta} V_1^i)(s^i, \mu) + (\epsilon^i + \xi + \epsilon^i \xi) ||Q^i(s^i, a^i, \mu, \nu)||_q - (\mathcal{B}^{\pi_\beta} V_2^i)(s^i, \mu) - (\epsilon^i + \xi + \epsilon^i \xi) ||Q^i(s^i, a^i, \mu, \nu)||_q|$$

$$= |(\mathcal{B}^{\pi_\beta} V_1^i)(s^i, \mu) - (\mathcal{B}^{\pi_\beta} V_2^i)(s^i, \mu)|.$$

Here, $||Q^i(s^i, a^i, \mu, \nu)||_q$ term cancels out each other since it is defined as the Q function under the benign transition, which can be well-learned in the benign mean-field game before the learning begins, and is thus not involved in the learning

process of $\mathcal{B}^R_{\epsilon^i,\xi} V^i(s^i, \mu, \epsilon^i, \xi)$. Thus, $\mathcal{B}^{\pi_\beta} V^i(s^i, \mu)$ is the transition under benign policy $\pi_\beta$. The Bellman operator $\mathcal{B}^{\pi_\beta} V^i$ can be written as:

$$\mathcal{B}^{\pi_\beta} V^i(s^i, \mu) = \sum_{a \in \mathcal{A}} \pi_\beta(a^i|s^i, \mu)\nu(a)[r + \gamma \sum_{s' \in \mathcal{S}} p(s'^i|s^i, a^i, \mu, \nu)V(s'^i, \mu')]$$

Thus, the proof proceeds by:

$$
\begin{aligned}
&|\mathcal{B}^R_{\epsilon^i,\xi} V_1^i(s^i, \mu, \epsilon^i, \xi) - \mathcal{B}^R_{\epsilon^i,\xi} V_2^i(s^i, \mu, \epsilon^i, \xi)| \\
=&|(\mathcal{B}^{\pi_\beta} V_1^i)(s^i, \mu) - (\mathcal{B}^{\pi_\beta} V_2^i)(s^i, \mu)|. \\
=&|\sum_{a \in \mathcal{A}} \pi_\beta(a^i|s^i, \mu)\nu(a)[r + \gamma \sum_{s' \in \mathcal{S}} p(s'^i|s^i, a^i, \mu, \nu)V_2^i(s'^i, \mu')] - \\
&\sum_{a \in \mathcal{A}} \pi_\beta(a^i|s^i, \mu)\nu(a)[r + \gamma \sum_{s' \in \mathcal{S}} p(s'^i|s^i, a^i, \mu, \nu)V_2^i(s'^i, \mu')]| \\
=&|\sum_{a \in \mathcal{A}} \pi_\beta(a^i|s^i, \mu)\nu(a)\gamma \sum_{s' \in \mathcal{S}} p(s'^i|s^i, a^i, \mu, \nu)(V_1^i(s'^i, \mu') - V_2^i(s'^i, \mu'))| \\
\leq&\gamma \sum_{a \in \mathcal{A}} \pi_\beta(a^i|s^i, \mu)\nu(a) \sum_{s' \in \mathcal{S}} p(s'^i|s^i, a^i, \mu, \nu)|V_1^i(s'^i, \mu') - V_2^i(s'^i, \mu')| \\
=&\gamma|V_1^i(s'^i, \mu') - V_2^i(s'^i, \mu')|
\end{aligned}
$$

This completes the proof. $\qquad\square$

### A.6. Proof for Proposition. 4.4

First, we apply first-order Taylor expansion to Q function under $\hat{\pi}$, $Q_{\hat{\pi}}^i(s^i, a^i, \mu, \nu)$. Similar to proof of A.4, we use $\hat{\pi}_\alpha^i = \hat{\pi}^i - \pi_\beta^i$ as a shorthand, resulting in:

$$Q_{\hat{\pi}}^i(s^i, a^i, \mu, \nu) = Q_{\pi_\beta}^i(s^i, a^i, \mu, \nu) + \min_{||\hat{\pi}_\alpha^i||_p \leq \epsilon^i, ||\hat{\nu}_\alpha||_p \leq \xi} \sum_{a^i, a \in \mathcal{A}} \hat{\pi}_\alpha \nu_\alpha Q_{\pi_\beta}^i(s^i, a^i, \mu, \nu).$$

Using Hölder's Inequality, we have:

$$\left|\left| \min_{||\hat{\pi}_\alpha^i||_p \leq \epsilon^i, ||\nu_\alpha||_p \leq \xi} \sum_{a^i, a \in \mathcal{A}} \hat{\pi}_\alpha \nu_\alpha Q_{\pi_\beta}^i(s^i, a^i, \mu, \nu) \right|\right|_1 \leq ||\hat{\pi}_\alpha^i||_p ||\nu_\alpha||_p ||Q_{\pi_\beta}^i(s^i, a^i, \mu, \nu)||_q,$$

with minimum achieved when $\hat{\pi}_\alpha^i$ and $\nu_\alpha$ aligns negatively with $Q_{\pi_\beta}^i(s^i, a^i, \mu, \nu)$, i.e.,

$$\hat{\pi}_\alpha^i \nu_\alpha = -\epsilon^i \xi \frac{Q_{\pi_\beta}^i(s^i, a^i, \mu, \nu)^{q-1}}{||Q_{\pi_\beta}^i(s^i, a^i, \mu, \nu)||_q^{q-1}}.$$

Using this worst-case $\hat{\pi}_\alpha^i \nu_\alpha$, we get:

$$
\begin{aligned}
\left|\left| \min_{||\hat{\pi}_\alpha^i||_p \leq \epsilon^i, ||\nu_\alpha||_p \leq \xi} \sum_{a^i, a \in \mathcal{A}} \hat{\pi}_\alpha \nu_\alpha Q_{\pi_\beta}^i(s^i, a^i, \mu, \nu) \right|\right|_1 &= ||\hat{\pi}_\alpha^i||_p ||\nu_\alpha||_p ||Q_{\pi_\beta}^i(s^i, a^i, \mu, \nu)||_q \\
&= \epsilon^i \xi ||Q_{\pi_\beta}^i(s^i, a^i, \mu, \nu)||_q
\end{aligned}
$$

We omit $\pi_\beta$ in $Q_{\pi_\beta}^i$ in our main text for conciseness. This completes the proof. $\qquad\square$

### A.7. Proof of Proposition. 4.5

We aim to prove that the optimal solution set $\mathcal{K}^*$ and the corresponding worst-case policy $\pi_\alpha^*$ of the original HAD-MFC $\mathcal{G} := \langle \mathcal{N}, \mathcal{S}, \mathcal{A}, \mathcal{P}, R, \mu_0, \nu_0, \gamma \rangle$ can be recovered by finding the optimal solution $\mathcal{K}_\mathcal{M}^*$ of the upper-level MDP $\mathcal{M} := \langle \mathcal{S}, \epsilon, \mathcal{N}, \tilde{\mathcal{P}}, \tilde{R}, \gamma \rangle$ and the exact solution to the lower-level regularized Bellman operator $\mathcal{B}^R_{\epsilon^i,\xi}$.

First, the lower-level problem requires calculating $\min_{\pi_\alpha} J(\pi_\alpha, \pi_\beta)$. In Proposition. 4.2, by Rockafellar-Fenchel transform, we have shown that:

$$
\begin{aligned}
\max_{\pi_\alpha} V^i(s^i,\mu) - (\hat{\mathcal{B}}^{\hat{\pi}} V^i)(s^i,\mu) &= V^i(s^i,\mu) - \mathcal{B}^R_{\epsilon^i,\xi} V^i(s^i,\mu,\epsilon^i,\xi) \\
&= V^i(s^i,\mu) - (\mathcal{B}^{\pi_\beta} V^i)(s^i,\mu) + (\epsilon^i + \xi + \epsilon^i\xi)\|Q^i(s^i,a^i,\mu,\nu)\|_q,
\end{aligned}
\tag{23}
$$

where $(\hat{\mathcal{B}}^{\hat{\pi}} V^i)(s^i,\mu)$ refers to the Bellman operator with the worst-case adversary. Rockafellar-Fenchel transform holds when the uncertainty set is convex, proper, and lower semi-continuous. This is satisfied by our rectangular p-norm bounded uncertainty set, as shown by the proof in Proposition. 4.1. Hence, the transform yields the exact optimal value of the original lower-level problem. Here, Fenchel-Rockafellar transform requires the convexity of the uncertainty set only, and do not require the value function or the policy to be convex.

Second, for the optimality of the upper-level MDP $\mathcal{M} := \langle \mathcal{S}, \epsilon, \mathcal{N}, \tilde{\mathcal{P}}, \tilde{R}, \gamma \rangle$, the reward in Eqn. 12 is defined on the optimal value of the lower-level problem. By Bellman's theorem (Bellman, 1966), there exists an optimal policy for the MDP that maximizes the expected cumulative reward. Thus, solving this MDP yields the optimal solution $\mathcal{K}^*_\mathcal{M} \subseteq \mathcal{N}$ for the upper-level, which is also the optimal solution $\mathcal{K}^* \subseteq \mathcal{N}$ of HAD-MFC.

Finally, for the lower-level problem, since the high-level vulnerable agent selection yields the same result, the lower-level problem face the mean-field MARL problem with same transition dynamics and same group of victims. Thus, the lower-level problem yields the same lower-level optimal policy $\pi_\alpha$.

Since both the lower-level transformation and the upper-level MDP mapping are exact (lossless), finding the optimal solution to the decomposed problem is mathematically equivalent to finding the optimal solution to the original HAD-MFC. $\qquad\square$

## B. Algorithm details

Our VAI algorithm involves a multi-step approach. In step 1, we evaluate the value function using regularized mean-field Bellman operator $\mathcal{B}^R_{\epsilon^i,\xi}$. In step 2, we solve the upper-level problem by training an RL algorithm to sequentially identify the most vulnerable agents, resulting in a set of vulnerable agents. Finally, in step 3, we train an adversarial policy on these identified vulnerable agents using MFC.

**Step 1.** In this step, we evaluate the value function $V^i(s^i,\mu,\epsilon^i,\xi)$ via regularized mean-field Bellman operator $\mathcal{B}^R_{\epsilon^i,\xi}$. The input is a set of trajectories sampled from a cooperative MFC policy. These trajectories are collected by performing 100 rollouts using the fixed, pre-trained cooperative policy $\pi_\beta$. Our pilot study shows adding additional rollouts do not yield better performance. Note that we assume shared $V^i$ and $Q^i$ for all agents.

**Step 2.** In this step, we train an RL agent to sequentially identify the most vulnerable agent. In our paper, we train this agent via Q-learning (Mnih et al., 2015b). Otherwise, we use a greedy algorithm to identify the most vulnerable agents. We hereby propose both our VAI-RL and VAI-greedy algorithm.

The third and final step is to solve the lower-level problem, i.e., train a zero-sum, worst-case adversarial policy on the selected set $\mathcal{K}$. This can be done by any standard MFC algorithm. In our algorithm, we use MF-AC (Yang et al., 2018) with shared reward as an example.

## C. Experiment details

### C.1. Environment details

We evaluate our algorithm on three environments: Magent, Vicsek, and Taxi. Visualizations of these environments are provided in Fig. 3.

**Magent.** The Magent platform (Zheng et al., 2018) supports large-scale multi-agent reinforcement learning. We test our algorithm on Battle task. In battle, agents engage in large-scale combat, earning rewards based on performance. We focus on the left-side agents for vulnerability identification.

**Taxi.** The Taxi supply-demand matching environment (Nguyen et al., 2018) allows agents to control taxis, receiving partial observations of their location and neighboring zones. A global reward is given for maintaining an optimal ratio of available taxis to demand in each zone.

---

**Algorithm 1** Step 1: computing value function $V^i(s^i, \mu, \epsilon^i, \xi)$.

---

**Input:** Trajectories $\tau_\beta$ sampled from cooperative policy $\pi_\beta$.
**Output:** Trained value function $V^i(s^i, \mu, \epsilon^i, \xi)$ via regularized mean-field Bellman operator $\mathcal{B}^R_{\epsilon^i, \xi}$.

  1: // Estimate $Q^i$ without perturbation
  2: **for** Iterations Iter = 0, 1, 2, ... K **do**
  3:      **for** Minibatch $\tau$ in trajectories $\tau_\beta$ **do**
  4:          Extract $[s, a, \mu, \nu, r, s', a', \mu', \nu']$ from $\tau$.
  5:          Compute $Q^i(s^i, a^i, \mu, \nu)$ and $Q^i(s'^i, a'^i, \mu', \nu')$
  6:          Update $Q^i$ by minimizing $(\gamma Q^i(s'^i, a'^i, \mu', \nu') + r - Q^i(s^i, a^i, \mu, \nu))^2$.
  7:      **end for**
  8: **end for**
  9: // Estimate $V^i$ with perturbation
10: **for** Iterations Iter = 0, 1, 2, ... K **do**
11:      **for** Minibatch $\tau$ in trajectories $\tau_\beta$ **do**
12:          **for** Sample $i$ = 0, 1, 2, ... B **do**
13:              Extract $[s^i, a^i, \mu, \nu, r, s'^i, a'^i, \mu', \nu']$ from $\tau_i$.
14:              Sample $\xi^i \sim Uniform[0, 2^{1/p}]$, $\epsilon^i \sim Bernoulli(\xi)$.
15:              Compute $V^i(s^i, \mu, \epsilon^i, \xi^i)$ and $V^i(s'^i, \mu', \epsilon^i, \xi^i)$.
16:              Compute $Q^i(s'^i, a'^i, \mu', \nu')$ using estimated $Q^i$ from $\tau_\beta$.
17:              Compute $\min_{||a'^i_\alpha||_p \leq \epsilon^i, ||\nu'_\alpha||_p \leq \xi^i} ||Q^i(s'^i, (a'^i_\beta + a'^i_\alpha), \mu', (\nu'_\beta + \nu'_\alpha))||_q$ using PGD (Madry, 2017),
18:              Update $V^i$ using Eqn. 13.
19:          **end for**
20:      **end for**
21: **end for**

---

**Algorithm 2** Step 2: Vulnerable Agent Identification, using Q learning (VAI-RL).

---

**Input:** Q function for vulnerable agent identification $Q(s, \epsilon, n)$, trained value function $V^i(s, \mu, \epsilon^i, \xi)$.
**Output:** Trained Q function for vulnerable agent identification $Q(s, \epsilon, n)$, set of vulnerable agents $\mathcal{K}$.

  1: Initialize vulnerable agent identification policy $Q(s, \epsilon, n)$, $\mathcal{K} = \varnothing$, $\epsilon_0 = \{0\}_N$.
  2: **for** Episode = 0, 1, 2, ... E **do**
  3:      **for** k = 1, 2, ... K **do**
  4:          Perform rollout under $n_k = \mathrm{argmax}_{n \in \mathcal{N}} Q^n(s^n_k, \epsilon^n_k, \xi_k)$, update $\epsilon_k, \xi_k$.
  5:          $\mathcal{K} \leftarrow \mathcal{K} \cup n_k$.
  6:          Compute $V^i(s, \mu, \epsilon^i, \xi)$ for all agents.
  7:          Compute reward $r_k = \frac{1}{N} \sum_{i \in \mathcal{N}} \left( V^i(s^i_0, \mu_0, \epsilon^i_{k-1}, \xi_{k-1}) - V^i(s^i_0, \mu_0, \epsilon^i_k, \xi_k) \right)$ following Eqn. 12.
  8:      **end for**
  9:      Save tuple $< s_k, \epsilon_k, n_k, r_k >$ in replay buffer.
10:      Update $Q(s, \epsilon, n)$ via DQN.
11: **end for**

---

**Vicsek.** The Vicsek model (Vicsek et al., 1995) simulates collective motion in flocks, where agents adjust their direction based on neighbors to maximize directional agreement. This environment operates in a continuous action space, where each agent selects a continuous angle to navigate.

### C.2. Implementation Details

Our implementation builds upon the mean-field MARL framework of Yang et al. (2018). We describe the key design choices for each baseline below, with an emphasis on ensuring the fairest possible comparison given each method's original formulation.

**Graph construction for graph-based baselines.** Our mean-field environments (Battle, Taxi, Vicsek) do not provide an explicit inter-agent graph structure, which is a prerequisite for graph-based methods such as PIANO. Rather than omitting these baselines entirely, we construct a heuristic adjacency matrix as follows: if agent $j$ falls within the observation range of

---

**Algorithm 3** Step 2: Vulnerable Agent Identification, using greedy algorithm (VAI-Greedy).

---

**Input:** Trained value function $V^i(s, \mu, \epsilon^i, \xi)$.
**Output:** Set of vulnerable agents $\mathcal{K}$.
1:  Initialize $\mathcal{K} = \varnothing$, $\epsilon_0 = \{0\}_N$.
2:  **for** k = 1, 2, ... K **do**
3:      **for** i = 1, 2, ... N **do**
4:          $r_k^{max} = -\infty$, $n_k^{max} = 1$.
5:          **if** $n_k^i \in \mathcal{K}$ **then**
6:              pass
7:          **else**
8:              Perform rollout under $n_k^i$, compute $\epsilon_k^i$, $\xi_k$.
9:              Compute $V^i(s, \mu, \epsilon^i, \xi)$ for all agents.
10:             Compute reward $r_k = \frac{1}{N} \sum_{i \in \mathcal{N}} \left( V^i(s_0^i, \mu_0, \epsilon_k^i, \xi_k) - V^i(s_0^i, \mu_0, \epsilon_{k-1}^i, \xi_{k-1}) \right)$ following Eqn. 12.
11:         **end if**
12:         **if** $r_k \geq r_k^{max}$ **then**
13:             $r_k^{max} \leftarrow r_k$, $n_k^{max} \leftarrow n_k^i$.
14:             $\mathcal{K} \leftarrow \mathcal{K} \cup n_k^i$.
15:         **end if**
16:     **end for**
17: **end for**

---

**Algorithm 4** Step 3: Learning the Adversarial Policy for Lower-Level Problem.

---

**Input:** Adversarial policy $\pi_\alpha$, victim policy $\pi_\beta$, set of vulnerable agents $\mathcal{K}$, perturbation budget $\epsilon^i$.
**Output:** Trained adversarial policy $\pi_\alpha$.
1:  Initialize vulnerable agent identification policy $\pi^{VAI}$, $\mathcal{K} = \varnothing$.
2:  **for** Episode = 0, 1, 2, ... E **do**
3:      **for** t = 1, 2, ... T **do**
4:          Get $s_t^i$, $\mu$ from environment.
5:          Compute $\hat{\pi}^i(a_t^i | s_t^i, \mu) = \epsilon^i \pi_\alpha^i(\cdot)(a_t^i | s_t^i, \mu) + (1 - \epsilon^i) \pi_\beta^i(a_t^i | s_t^i, \mu)$ for all $i \in \mathcal{N}$.
6:          Sample $a_t^i \sim \hat{\pi}^i(\cdot | s_t^i, \mu)$. Compute $\nu$.
7:          Calculate $r_t$ from environment.
8:          Store $[s_t^i, a_t^i, \mu_t, \nu_t, r_t]$ in trajectory $\tau$.
9:      **end for**
10:     **for** k = 0, 1, ... K **do**
11:         Sample a batch $\tau_k$ from trajectory $\tau$.
12:         Update the critic $Q^i(s_i, a_i, \mu, \nu)$ by minimizing TD loss $(\gamma Q^i(s'^i, a'^i, \mu', \nu') + r - Q^i(s^i, a^i, \mu, \nu))^2$.
13:         Update the policy of adversary $\pi_\alpha$ by sampled policy gradient $-\nabla_{\pi_\alpha} \log \pi_\alpha Q^i(s^i, a^i, \mu, \nu)$.
14:     **end for**
15: **end for**

---

agent $i$, we place an undirected edge between them with weight 1. This gives graph-based baselines the richest structural information available in our setting, and ensures they are not unfairly handicapped by a trivially uninformative graph.

**PIANO.** In the original PIANO formulation (Li et al., 2022), the upper-level selector receives a reward signal directly from the environment immediately after agent selection, without requiring any lower-level policy training. In our HAD-MFC setting, obtaining a faithful upper-level reward requires training a full lower-level adversarial policy for each candidate group, a process that takes several hours per group and renders exhaustive upper-level search computationally infeasible. This intractability is precisely the core difficulty that Proposition 4.2 addresses, by replacing expensive lower-level rollouts with a closed-form value proxy. For PIANO specifically, we approximate the upper-level reward during selection using the performance of lower-level attackers executing a *random* policy, which is the closest analogue to PIANO's original reward structure that does not require full lower-level training. After the final group is selected, we train a complete adversarial policy on that group to obtain the reported attack performance. We believe this constitutes the fairest adaptation of PIANO

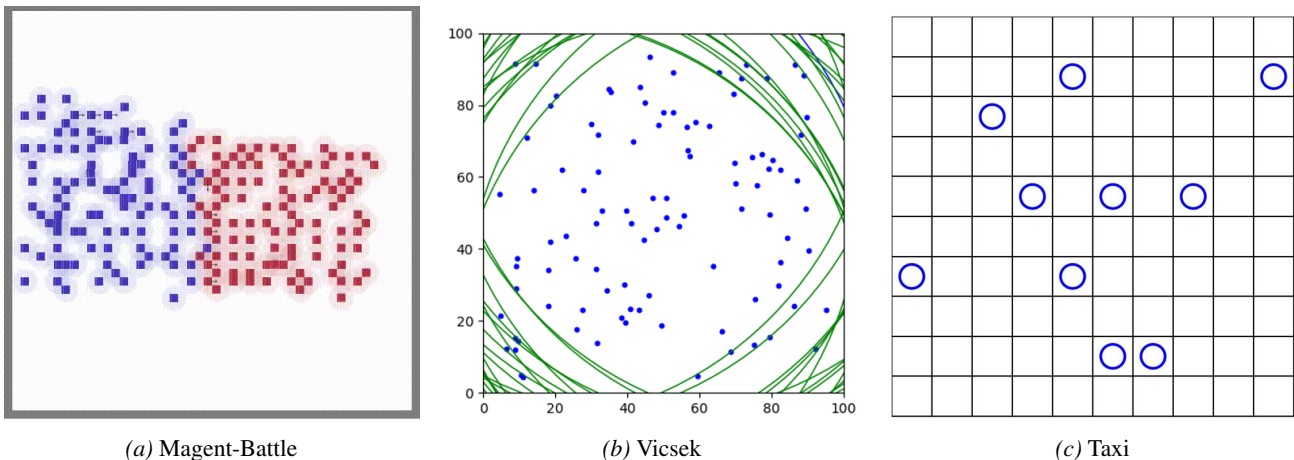

*(a)* Magent-Battle        *(b)* Vicsek        *(c)* Taxi

*Figure 3.* Environments used in our experiments. The task Battle in Magent are proposed in Zheng et al. (2018). The Taxi environment follows Nguyen et al. (2018). Vicsek model follows the dynamics defined by Vicsek et al. (1995).

to our setting, and note that the comparison therefore understates PIANO's advantage relative to a hypothetical oracle that could evaluate the true lower-level reward for every candidate group.

**Bi-Level RL.** Both the input and output structures of Bi-Level RL match those of VAI-RL. The key difference is that Bi-Level RL provides the upper-level reward only after all $K$ agents have been selected and the full lower-level adversarial policy has been trained, resulting in a sparse reward signal. The upper-level and lower-level attackers share the same cumulative reward.

**RTCA.** We follow the original RTCA methodology (Zhou & Liu, 2023) and tune its hyperparameters for best performance in our environments.

Detailed descriptions of our VAI method and baseline implementations are provided in Appendix B, with hyperparameter settings in Appendix C.3.

### C.3. Hyperparameters

In this section, we list all hyperparameters used Battle, Taxi and Vicsek environment. All hyperparameters are shared by VAI and other baselines. The hyperparameters used by Battle is at Table. 2.

*Table 2.* Hyperparameters for Battle environment.

| Hyperparameter | Value | Hyperparameter | Value |
| --- | --- | --- | --- |
| agent | 64/144 | mapsize | 40/60 |
| adv. num | 12.5/25/50% | save_every | 5 |
| n_round | 2000/5000 | maxsteps | 400 |
| gamma | 0.95 | lr | 1e-4 |
| tau | 0.005 | batch_size | 64 |
| memory_size | 80000 | | |

The hyperparameters used by Taxi is at Table. 3.

The hyperparameters used by Vicsek is at Table. 4.

## D. Additional Result

### D.1. Complexity and Runtime Efficiency

**Hardware Specifications:** We note that the runtime evaluations for the Taxi and Vicsek environments were conducted on a different hardware configuration (NVIDIA L40S GPU and Intel Xeon Gold 6448Y) compared to the Battle environment (NVIDIA RTX 3090 GPU and AMD EPYC 7H12 64-Core Processor) due to resource availability during the rebuttal phase.

*Table 3.* Hyperparameters for Taxi environment.

| Hyperparameter | Value | Hyperparameter | Value |
|---|---|---|---|
| optimizer | Adam | number of agents $N$ | 50/100 |
| PPO clip | 0.2 | hidden dim | 0.99 |
| number of adv agents $M$ | 4/16/36 | critic loss coefficient $c_1$ | 0.5 |
| map size | $10 \times 10$ | maximum number of policy training episodes | $2 \cdot 10^6$ |
| entropy loss coefficient $c_2$ | 0.01 | data_chunk_length | 10 |
| entropy_coef | 0.01 | eorder num | 100 |
| actor learning rate | $3 \cdot 10^{-5}$ | discount factor $\gamma$ | 0.99 |
| gae_lambda | 0.95 | gain | 0.01 |
| gamma | 0.99 | hidden_sizes | [128, 128] |
| order price | 1/2 | update every $E$ episodes | 5 |
| batch size | 64 | length of an episode $T$ | 20 |

*Table 4.* Hyperparameters for Vicsek environment.

| Hyperparameter | Value | Hyperparameter | Value |
|---|---|---|---|
| action_aggregation | prod | activation_func | relu |
| actor_num_mini_batch | 1 | clip_param | 0.05 |
| critic_epoch | 5 | critic_lr | 0.0005 |
| critic_num_mini_batch | 1 | cuda | true |
| cuda_deterministic | true | data_chunk_length | 10 |
| entropy_coef | 0.01 | episode_length | 200 |
| eval_episodes | 20 | eval_interval | 25 |
| gae_lambda | 0.95 | gain | 0.01 |
| gamma | 0.99 | hidden_sizes | [128, 128] |
| huber_delta | 10.0 | initialization_method | orthogonal_ |
| world_size | 100 | log_interval | 5 |
| lr | 0.0005 | max_grad_norm | 10.0 |
| torus | true | n_eval_rollout_threads | 10 |
| n_rollout_threads | 5 | num_env_steps | 5000000 |
| opti_eps | 1e-05 | ppo_epoch | 5 |
| recurrent_n | 1 | render_episodes | 10 |
| seed | 1 | seed_specify | true |
| share_param | true | std_x_coef | 1 |
| std_y_coef | 0.5 | torch_threads | 4 |
| use_clipped_value_loss | true | use_eval | true |
| use_feature_normalization | true | use_gae | true |
| use_huber_loss | true | use_linear_lr_decay | false |
| use_max_grad_norm | true | use_policy_active_masks | true |
| use_popart | true | use_proper_time_limits | true |
| use_recurrent_policy | false | use_render | false |
| value_loss_coef | 1 | weight_decay | 0 |
| use_agent_states_init | true | bearing_bins | 8 |
| comm_radius | 20 | distance_bins | 8 |
| dynamics | unicycle | nr_agents | 100 |
| obs_mode | fix_acc | | |

However, all methods within any specific environment were evaluated strictly on the exact same hardware to ensure a fair relative comparison.

In this section, we analyze the computation cost of VAI and baselines in Battle environment. The overall results are shown in Table. 5. The runtimes are averaged across 5 runs and we do not find large variations between different runs. Our VAI consists of two stages. First, we train a value function using the regularized mean-field Bellman operator (Proposition 4.2).

*Table 5.* Runtime comparison of all methods in Battle environment (in hours). VAI requires a one-time training of a regularized value function (Proposition 4.2), which takes approximately 1 hour and is reused across all VAI variants and adversarial-agent settings. This one-time cost is not included in the table, as it is amortized across all experiments.

| Agent Num | Adv Num | Random | DC | Bi-Level RL | PIANO | RTCA | VAI-Greedy | VAI-RL |
|---|---|---|---|---|---|---|---|---|
| 64 | 8 | 1.40 ± 0.13 | 1.38 ± 0.07 | 1.50 ± 0.23 | 1.49 ± 0.20 | 1.71 ± 0.16 | 1.36 ± 0.15 | 1.42 ± 0.17 |
|  | 16 | 1.41 ± 0.13 | 1.34 ± 0.10 | 1.49 ± 0.16 | 1.69 ± 0.24 | 2.17 ± 0.07 | 1.44 ± 0.88 | 1.43 ± 0.20 |
|  | 32 | 1.65 ± 0.16 | 1.60 ± 0.12 | 1.92 ± 0.18 | 1.85 ± 0.23 | 2.88 ± 0.18 | 1.66 ± 0.57 | 1.78 ± 0.35 |
| 144 | 18 | 1.24 ± 0.10 | 1.22 ± 0.08 | 1.29 ± 0.14 | 1.56 ± 0.05 | 1.54 ± 0.20 | 1.24 ± 0.84 | 1.38 ± 0.18 |
|  | 36 | 1.50 ± 0.11 | 1.48 ± 0.04 | 1.53 ± 0.15 | 1.77 ± 0.21 | 2.40 ± 0.36 | 1.56 ± 0.91 | 1.58 ± 0.26 |
|  | 72 | 3.76 ± 0.12 | 3.62 ± 0.09 | 4.02 ± 0.21 | 3.98 ± 0.26 | 5.54 ± 0.44 | 3.93 ± 0.68 | 4.15 ± 0.85 |

*Table 6.* Runtime comparison of different methods in Taxi environment (in hours). VAI requires a one-time training of a regularized value function (Proposition 4.2), which takes approximately 1 hour and is reused across all VAI variants and adversarial-agent settings. This one-time cost is not included in the table, as it is amortized across all experiments.

| Agent Num | Adv Num | Random | DC | VAI-Greedy | VAI-RL |
|---|---|---|---|---|---|
| 50 | 4 | 0.5341 ± 0.0002 | 0.5373 ± 0.0002 | 0.6731 ± 0.1540 | 0.7903 ± 0.1124 |
|  | 16 | 0.5344 ± 0.0003 | 0.5377 ± 0.0002 | 0.8249 ± 0.1529 | 1.0214 ± 0.1571 |
|  | 36 | 0.5349 ± 0.0001 | 0.5382 ± 0.0002 | 0.9594 ± 0.1554 | 1.2193 ± 0.1788 |
| 100 | 4 | 0.5387 ± 0.0002 | 0.5515 ± 0.0001 | 0.7357 ± 0.1472 | 1.2372 ± 0.1264 |
|  | 16 | 0.5390 ± 0.0003 | 0.5529 ± 0.0021 | 1.0782 ± 0.1490 | 1.3491 ± 0.1127 |
|  | 36 | 0.5395 ± 0.0001 | 0.5526 ± 0.0005 | 1.5407 ± 0.1452 | 2.5340 ± 0.1574 |

This process, which involves computing a regularization Q function, is comparable in complexity to a typical value update in mean-field MARL and takes about 1 hour. Once trained, the value function is fixed and reused across both VAI-Greedy and VAI-RL, so it is trained only once.

We find the overall computation time to be tractable, with most tasks requiring roughly one hour of training. A notable exception is the case of 144 agents with 72 adversaries, which requires 3–5 hours. This increased cost arises because the large number of adversaries introduces substantial CPU and GPU bottlenecks. However, this slowdown is inherent to current mean-field MARL frameworks rather than specific to our approach, and the additional training time affects both our method and all baselines equally.

For the baselines, the computation cost varies according to how they select adversaries. Random and DC rely on simple heuristics and therefore incur negligible overhead—their runtime is dominated by training the underlying RL policy. However, their performance is often limited because they ignore task dynamics. Bi-Level RL introduces moderate additional cost by training a separate high-level selector, and PIANO incurs a similar overhead due to its GNN-based selector. RTCA is the most computationally expensive baseline, as it maintains 10 evolutionary populations for adversary selection, resulting in significantly higher runtime.

To compare, our VAI-Greedy performs adversary selection by ranking agent vulnerabilities using the value function. It has O(NK) complexity (selecting K agents from N), but incurs negligible cost in practice (<1 second for both 64 and 144 agents) since it does not need require additional training. VAI-RL uses Q-learning to sequentially select vulnerable agents. It incurs modest computation overhead compared to baselines. Our VAI-RL is fast since it does not need interactions with the environment. Its complexity scales as O(K) (number of adversaries), and can be efficiently extended using techniques to handle large numer of agents by using large-action space approximation techniques in RL (Dulac-Arnold et al., 2015).

Overall, the computation cost of all baselines is manageable, except in settings with a very large number of adversaries, where the underlying mean-field MARL training becomes the dominant bottleneck. For our method, although VAI requires a one-time value-function training stage, this cost is amortized across both VAI-Greedy and VAI-RL, as well as different number of adversaries. Beyond this initialization, VAI's selection procedures are highly efficient, achieving runtime

*Table 7.* Runtime comparison of different methods in Vicsek environment (in hours). VAI requires a one-time training of a regularized value function (Proposition 4.2), which takes approximately 1 hour and is reused across all VAI variants and adversarial-agent settings. This one-time cost is not included in the table, as it is amortized across all experiments.

| Agent Num | Adv Num | Random | DC | VAI-Greedy | VAI-RL |
|---|---|---|---|---|---|
| 100 | 20 | 2.0043 ± 0.0003 | 2.0165 ± 0.0002 | 2.6142 ± 0.1624 | 3.1205 ± 0.1245 |
|  | 35 | 2.0071 ± 0.0005 | 2.0184 ± 0.0004 | 3.1258 ± 0.1587 | 3.9842 ± 0.1412 |
|  | 50 | 2.0108 ± 0.0002 | 2.0217 ± 0.0003 | 3.8471 ± 0.1511 | 4.7567 ± 0.1790 |
| 400 | 80 | 2.0256 ± 0.0004 | 2.0521 ± 0.0005 | 3.9102 ± 0.1402 | 5.8821 ± 0.1311 |
|  | 140 | 2.0288 ± 0.0006 | 2.0584 ± 0.0018 | 5.1856 ± 0.1524 | 7.4242 ± 0.1187 |
|  | 200 | 2.0314 ± 0.0003 | 2.0567 ± 0.0006 | 7.2248 ± 0.1498 | 9.8241 ± 0.1625 |

*Table 8.* Performance comparison on Battle environment under different perturbation budgets $\epsilon$ ($\downarrow$).

| Num Agent | Adv Agent | Method | $\epsilon$ | | |
|---|---|---|---|---|---|
| | | | 0.3 | 0.5 | 0.7 |
| 64 | 32 | Random | $191.70_{\pm 72.59}$ | $198.50_{\pm 96.22}$ | $167.40_{\pm 74.31}$ |
| | | DC | $184.90_{\pm 52.89}$ | $155.30_{\pm 66.64}$ | $137.70_{\pm 55.81}$ |
| | | Bi-Level RL | $116.10_{\pm 62.95}$ | $103.80_{\pm 34.74}$ | $95.37_{\pm 33.17}$ |
| | | PIANO | $107.25_{\pm 14.09}$ | $94.93_{\pm 24.25}$ | $77.85_{\pm 14.74}$ |
| | | RTCA | $133.80_{\pm 25.80}$ | $103.2_{\pm 13.64}$ | $86.91_{\pm 24.53}$ |
| | | VAI-Greedy | $\mathbf{92.30_{\pm 5.53}}$ | $\mathbf{83.00_{\pm 29.51}}$ | $\mathbf{74.37_{\pm 26.34}}$ |
| | | VAI-RL | $\mathbf{78.95_{\pm 9.34}}$ | $\mathbf{78.35_{\pm 12.77}}$ | $\mathbf{55.29_{\pm 2.49}}$ |
| 144 | 72 | Random | $276.90_{\pm 54.72}$ | $92.00_{\pm 66.94}$ | $-115.80_{\pm 9.52}$ |
| | | DC | $-77.05_{\pm 32.69}$ | $-109.50_{\pm 21.53}$ | $-229.90_{\pm 2.02}$ |
| | | Bi-Level RL | $-62.74_{\pm 29.43}$ | $-93.40_{\pm 32.32}$ | $-252.20_{\pm 7.81}$ |
| | | PIANO | $\mathbf{-88.48_{\pm 14.72}}$ | $-93.00_{\pm 27.09}$ | $-207.30_{\pm 5.03}$ |
| | | RTCA | $66.90_{\pm 42.23}$ | $-138.03_{\pm 6.94}$ | $-298.10_{\pm 3.43}$ |
| | | VAI-Greedy | $40.30_{\pm 8.54}$ | $\mathbf{-149.90_{\pm 10.61}}$ | $\mathbf{-338.60_{\pm 5.87}}$ |
| | | VAI-RL | $\mathbf{-151.60_{\pm 23.19}}$ | $\mathbf{-243.80_{\pm 27.42}}$ | $\mathbf{-406.30_{\pm 1.56}}$ |

*Table 9.* Performance comparison on Taxi environment under different perturbation budgets $\epsilon$ ($\downarrow$).

| Num Agent | Adv Agent | Method | $\epsilon$ | | | |
|---|---|---|---|---|---|---|
| | | | 0.3 | 0.5 | 0.7 | 1.0 |
| 50 | 25 | Random | $483.87_{\pm 15.94}$ | $461.43_{\pm 15.73}$ | $449.31_{\pm 63.91}$ | $503.39_{\pm 30.51}$ |
| | | DC | $486.54_{\pm 16.16}$ | $462.05_{\pm 12.62}$ | $380.65_{\pm 25.97}$ | $489.82_{\pm 24.81}$ |
| | | Bi-Level RL | $477.31_{\pm 13.65}$ | $467.28_{\pm 10.74}$ | $490.81_{\pm 19.34}$ | $508.67_{\pm 16.80}$ |
| | | PIANO | $472.27_{\pm 13.14}$ | $458.75_{\pm 12.49}$ | $430.56_{\pm 17.56}$ | $441.90_{\pm 21.16}$ |
| | | RTCA | $482.20_{\pm 14.69}$ | $\mathbf{428.63_{\pm 13.65}}$ | $392.85_{\pm 19.80}$ | $434.28_{\pm 19.51}$ |
| | | VAI-Greedy | $\mathbf{461.58_{\pm 15.74}}$ | $429.97_{\pm 12.82}$ | $\mathbf{331.20_{\pm 20.72}}$ | $\mathbf{321.01_{\pm 11.61}}$ |
| | | VAI-RL | $\mathbf{453.29_{\pm 14.48}}$ | $\mathbf{406.75_{\pm 13.39}}$ | $352.21_{\pm 17.12}$ | $\mathbf{299.82_{\pm 21.05}}$ |
| 100 | 50 | Random | $799.87_{\pm 40.42}$ | $792.99_{\pm 26.01}$ | $859.00_{\pm 53.35}$ | $938.78_{\pm 34.97}$ |
| | | DC | $807.40_{\pm 26.96}$ | $854.25_{\pm 19.72}$ | $835.10_{\pm 27.92}$ | $859.64_{\pm 63.08}$ |
| | | Bi-Level RL | $798.31_{\pm 20.79}$ | $771.67_{\pm 18.90}$ | $800.48_{\pm 44.75}$ | $817.33_{\pm 65.84}$ |
| | | PIANO | $797.20_{\pm 24.27}$ | $787.63_{\pm 23.95}$ | $798.18_{\pm 32.18}$ | $723.80_{\pm 152.61}$ |
| | | RTCA | $811.34_{\pm 22.13}$ | $781.21_{\pm 16.61}$ | $761.81_{\pm 48.46}$ | $868.56_{\pm 35.74}$ |
| | | VAI-Greedy | $\mathbf{778.59_{\pm 26.41}}$ | $\mathbf{750.75_{\pm 23.71}}$ | $\mathbf{722.62_{\pm 24.60}}$ | $\mathbf{605.63_{\pm 38.83}}$ |
| | | VAI-RL | $\mathbf{756.91_{\pm 12.71}}$ | $\mathbf{721.64_{\pm 15.51}}$ | $\mathbf{710.65_{\pm 29.07}}$ | $\mathbf{572.03_{\pm 25.05}}$ |

comparable to, or even lower than several baselines (particularly RTCA). This makes VAI a practical and scalable solution for large-scale multi-agent systems.

### D.2. Performance with Partial Perturbation Budgets $\epsilon$

While our main experiments examined the extreme setting of $\epsilon = 1$, where attackers fully control compromised agents, we additionally evaluate partial perturbations with $\epsilon = \{0.3, 0.5, 0.7\}$. For Battle, we use configurations of 64 agents with 32 adversaries and 144 agents with 72 adversaries (a 50% adversary ratio). We match this ratio in the Taxi environment with settings of 50 agents with 25 adversaries and 100 agents with 50 adversaries, and we also include $\epsilon = 1$ for Taxi since it was not covered in the main paper. All experiments are averaged across 5 random seeds.

As shown in Tables 8 and 9, both VAI-RL and VAI-Greedy generally outperform all baselines. These improvements are statistically significant under the nonparametric Friedman test (VAI-Greedy: $p < .005$, VAI-RL: $p < 0.05$) and Taxi (VAI-Greedy: $p < 0.05$, VAI-RL: $p < 0.05$). Several discussions are highlighted below.

First, VAI-RL outperforms VAI-Greedy in 13 of 14 settings, consistent with our main results showing that VAI-RL is more effective when many adversaries are present. These scenarios require finer-grained exploration of system vulnerabilities, where VAI-Greedy's simple selection process becomes less optimal. In contrast, VAI-RL better captures synergistic interactions among adversaries and outperforms VAI-Greedy under smaller perturbation budgets.

Second, both VAI-RL and VAI-Greedy consistently outperform all baselines across different perturbation budgets $\epsilon$. Although smaller $\epsilon$ naturally weakens adversarial impact, the advantage of VAI remains robust even under these more constrained conditions.

