# OpenReview forum: "Vulnerable Agent Identification in Large-Scale Multi-Agent Reinforcement Learning"
_ICML.cc/2026/Conference — ICML 2026 regular_

### Official Review · Reviewer_iAor · 2026-02-26

**Soundness:** 3
**Presentation:** 3
**Significance:** 3
**Originality:** 3
**Overall Recommendation:** 4
**Confidence:** 4

**Summary:**

The paper proposes a reinforcement learning (RL) method to determine subsets of vulnerable agents in a large-scale multi-agent systems (focusing on cooperative multi-agent RL settings), where vulnerability is measured by the performance degradation through adversarial agent replacements/manipulations. Due to the intractability of conventional black box multi-agent RL techniques, the paper proposes a hierarchical adversarial learning approach to (1) select a subset of $K$ vulnerable agents and (2) train an adversarial policy for that subset. Step (1) is achieved via RL, where the subset selection is framed as an MDP for $K$ steps, while step (2) is achieved via RL using a regularized mean-field operator via the Fenchel-Rockafellar transform of the originally learned value function. The approach, called VAI (Vulnerable Agent Identification), is evaluated in three benchmark domains and compared with rule-based and randomised approaches to selecting subsets of vulnerable agents.

**Compliance With Llm Reviewing Policy:**

Affirmed.

**Final Justification:**

After the rebuttal, my assessment remains the same.

**Key Questions For Authors:**

1. The paper assumes a fixed number $K$ of adversaries, which may not be known a priori. Is there any intuition or idea how this number could be determined in real-world settings?
2. The MDP formulation assumes the sequential selection of the $K$ most vulnerable agents. While the formulation (including the dense rewards) leads to tractable RL, the actual complexity/NP-hardness still remains, since the RL algorithm has to explore $K!$ possibilities in the worst-case, correct? This should be mentioned in a sentence in 4.2.

**Limitations:**

Yes

**Strengths And Weaknesses:**

**Soundness**:

The paper is technically sound. I have not verified all the proofs in the appendices, however the proof sketches provided in the main paper are clear and intuitive, and the assumptions are listed completely.

**Presentation**:

The paper is well-written and easy to follow. I appreciate the proof sketches, which support the reading flow and self-containment of the main paper.

The font sizes in Figure 1 are very small and I am not sure about the necessity of the Taxi example in the main paper. To save some space, I suggest moving it to the appendix in order to make the remaining plots larger.

**Significance**:

The contributions are significant and important to the multi-agent RL community, which is supported by the extensive literature coverage, theoretical analyses, and the experimental results.

**Originality**:

I consider the high-level approach as incremental, since high-level grouping and low-level training has already been explored for large-scale multi-agent RL settings, e.g., using meta-policy learning which is similar to the MDP-approach of the paper [1,2,3]. However, the low-level approach, i.e., the use of Fenchel-Rockafellar transform is intriguing.

**Literature**

[1] Phan et al., "VAST: Value Function Factorization with Variable Agent Sub-Teams", NeurIPS 2021

[2] Zang et al., "Automatic Grouping for Efficient Cooperative Multi-Agent Reinforcement Learning", NeurIPS 2023

[3] Li et al., "GTDE: Grouped Training with Decentralized Execution for Multi-agent Actor-Critic", AAAI 2025

---

> ### Author Rebuttal · Authors · 2026-03-29
>
> ### Official Response of Submission 14051 to Reviewer iAor
>
> We thank Reviewer iAor for appreciating the clarity, significant contribution and originality of our paper. A detailed response is provided below.
>
> ## Response
>
> > Q1: I consider the high-level approach as incremental, since high-level grouping and low-level training has already been explored for large-scale multi-agent RL settings, e.g., using meta-policy learning which is similar to the MDP-approach of the paper [1,2,3]. However, the low-level approach, i.e., the use of Fenchel-Rockafellar transform is intriguing.
>
> Please allow us to clarify that our method solves a different problem comparing with [1-3]. .
>
> > Q2: The font sizes in Figure 1 are very small and I am not sure about the necessity of the Taxi example in the main paper. To save some space, I suggest moving it to the appendix in order to make the remaining plots larger.
>
> Thanks for the advice. We will modify the organization accordingly in our final version.
>
> > Q3: The paper assumes a fixed number of adversaries, which may not be known a priori. Is there any intuition or idea how this number could be determined in real-world settings?
>
> Thanks for mentioning this. If adversaries take control of the agent and take non-cooperative actions, we can spot them from their actions. [1] examined this idea in MARL (not mean-field setting) which use Bayesian inference to detect whether an agent is compromised, and the detection result is used for further defense. If we have to estimate this a priori, we have to rely on the task record and estimate it in a case-specific way. For example, in a power grid, we can look at the history and esimate the percentage of nodes going down at the same time.
>
> > Q4: The MDP formulation assumes the sequential selection of the K most vulnerable agents. While the formulation (including the dense rewards) leads to tractable RL, the actual complexity/NP-hardness still remains, since the RL algorithm has to explore possibilities in the worst-case, correct? This should be mentioned in a sentence in 4.2.
>
> Thanks for mentioning this matter. Indeed the NP-hardness remains and RL serves as an approximation to this problem. We will make this clear in section 4.2.

---

> > ### Author Rebuttal · Reviewer_iAor · 2026-04-03
> >
> > Q2-Q4 are fine, but I am not happy with the answer to Q1, where I was referring to the *method* instead of the *problem*.
> >
> > Since I consider the mere application of the same/similar method to a different problem, which is still MARL, as incremental, I am keeping my score. For a raise, I need more methodical engagement.

---

> > > ### Author Response · Authors · 2026-04-03
> > >
> > > Please allow us to clarify that our method also use different method comparing with [1-3]. In our setting and [1-3] there both exists a grouping problem and a lower-level RL learning problem. While both level are tractable, the correlation between these levels make the problem hard to solve. There exists several fundamental differences.
> > >
> > > **We are not in cooperative setting and the methods in [1-3] may not apply**. [1-3] occurs in fully cooperative setting. In this way, the learned value functions reflects the actual value expected to gain. In our setting, however, we assume worst-case adversaries to minimize team reward. In this way, (1) the cooperative premise in [1-3] do not hold and individual agent values can not be combined monotonically, which violates the assumption of QMIX-style methods. (2) cooperation effectiveness do not reflect its vulnerability under attack. So, we have to learn another worst-case value estimate for the impact of different combinations being selected as adversary. This is done by our Fenchel-Rockafellar transform, which is our main contribution as stated in Proposition 4.2.
> > >
> > > **Due to our tractable worst-case value estimate, we know the result of grouping, which [1-3] lacks**. Since the value function $V(s, a, \mu, \nu)$ learned in Proposition 4.2 can estimate the attack result under a group of selected adversary, we can optimize on $V(s, a, \mu, \nu)$ directly. That is to say, we know the effect of current selected group of agent before actual attack! As such, for selecting K agents from N agents, we can do this by iteratively selecting the most vulnerable one! In contrast, [3]'s classification-based grouping is scalable, they do not give an estimate of how good current grouping is, and the upper and lower levels are still coupled. This is similar to our baseline Bi-level optimization, which solves the coupled upper- and lower- level problem together. Experiment results show that due to the complexity of our problem, the problem cannot be solved simply by bi-level optimization itself.
> > >
> > > However, it is worth acknowledging that the VAI-RL and VAI-Greedy algorithm is not the primary novelty. **Our key contribution key contribution is the derivation of a tractable adversarial value function $V(s, a, \mu, \nu)$ via Fenchel-Rockafellar duality (Proposition 4.2), which enables this RL and Greedy procedure to work correctly**. In cooperative methods [1–3], an analogous greedy strategy would fail because their individual sub-values $Q^i$ are not guaranteed to reflect true marginal contributions under adversarial perturbations. Our value function provides this guarantee; theirs do not.

---

### Official Review · Reviewer_ybT6 · 2026-03-09

**Soundness:** 2
**Presentation:** 3
**Significance:** 3
**Originality:** 2
**Overall Recommendation:** 4
**Confidence:** 4

**Summary:**

The paper studies vulnerable agent identification (VAI) in large-scale MARL, specifically, finding the k agents whose adversarial compromise causes the worst system degradation. It formulates this as a hierarchical bi-level problem (Hierarchical Adversarial Decentralized Mean Field Control ), shows the upper level is NP-hard, and proposes a tractable decomposition. The core contribution is a regularized mean-field Bellman operator that estimates worst-case value without requiring the expensive inner RL training loop. The upper-level combinatorial problem is then solved greedily or via RL. Experiments on Battle, Taxi, and Vicsek show outperformance over baselines.

**Compliance With Llm Reviewing Policy:**

Affirmed.

**Final Justification:**

Improvements are made.

**Key Questions For Authors:**

1. A formal argument showing the mean-field approximation remains valid in this adversarial setting is needed.
2. Position 4.2 and 4.3 require treating Q as a fixed coefficient, but Algorithm 1 learns Q and V simultaneously via TD updates. It is needed to clarify whether the converged solution of Algorithm 1 corresponds to the fixed point guaranteed by Proposition 4.2.
3. Proposition 4.5 claims the decomposition is optimality-preserving, yet the upper-level problem is NP-hard by Proposition 3.3. Specify what optimality-preserving means here and provide the approximation guarantees of VAI-Greedy and VAI-RL.
4. Clarify what metric is reported in Table 1, and provide statistical significance tests for the reported improvements given the substantially higher variance of VAI in several scenarios.

**Limitations:**

Yes

**Strengths And Weaknesses:**

Strengths:
- Identifying the vulnerable agents in the large population MARL setting, in this case, to the mean-field limit, is interesting.
- The use of Fenchel-Rockafellar duality to decouple the hierarchical structure is technically creative. When it works as claimed, it avoids training a separate adversarial policy for each candidate agent set, which would be completely intractable.

Weakness:
- The first concern is whether the mean-field approximation remains valid under adversarial attack. The framework first selects agents specifically to maximally disrupt the mean field, but the entire theoretical machinery assumes the mean-field approximation holds. Note that the original mean-field RL work by Yang et al. (2018) handles heterogeneous agents under cooperative/competitive play, not adversarial perturbation designed to distort the mean field. So there is a gap, especially when the adversary ratio is large.
- Another concern is with the fixed Q-function assumption. The derivation of the regularized mean-field Bellman operator in Prop. 4.2 and the contraction guarantee in Prop. 4.3 both require treating Q as a fixed coefficient during the Fenchel-Rockafellar transform. However, in Algorithm 1, Q is simultaneously being learned via TD updates while the regularized V is being trained using that same evolving Q. The paper does not discuss whether the converged solution of Algorithm 1 corresponds to the fixed point of the theoretical operator, i.e., whether the value function learned in practice is actually the one guaranteed by Proposition 4.2.
- It is also unclear whether optimality-preserving claimed in Prop. 4.5 refers to the equivalence of the decomposed formulation to the original HAD-MFC, or whether the proposed algorithms are claimed to solve it exactly, knowing that upper level problem is NP-hard. In addition, the approximation guarantees of VAI-Greedy and VAI-RL are not discussed.
- The runtime and computational scaling analysis in Appendix D.1 is limited to the Battle environment only. It is unclear whether the computational conclusions generalize to Taxi and Vicsek, which have different dynamics and state-action structures.
- The paper just oscillates between positioning VAI as an attack method and a defense diagnostic tool. It is better to unify them to have a consistence presentation.

- What is the metric plotted in Table 1? Additionally, for several scenarios, the mean of VAI methods is better than baselines but with substantially higher variance, which raises questions about the statistical significance of the reported improvements. Presenting the results visually would make it easier to assess whether the advantages are consistent or driven by occasional outliers.
- Too many typos: For example, The coupled nature characterize the complexity issue of our problem; the property do not require the value function or policy to be convex; our method requires assess to system trajectories.

---

> ### Author Rebuttal · Authors · 2026-03-29
>
> ### Official Response of Submission 11855 to Reviewer ybT6
>
> We thank Reviewer ybT6 for handling our paper. A detailed response is provided below.
>
> ## Response
>
> > Q1: Validity of mean-field approximation under adversarial attack.
>
> We respectfully disagree that there is a gap in the mean-field approximation. First, we assume the victim policy is well-trained and in its deployment phase. As such, the **victim operates with a fixed policy, is unaware of potential adversaries, and always aggregates the actions of all agents into a single mean field**. Second, the theory established by Yang et al. (2018) assumes only agent exchangeability and the large-population limit, neither of which is violated by our adversarial setup. Moreover, their theory is grounded in stochastic games, which naturally accommodate agents with different objectives. In our setting, the attacker influences the mean field solely through its actions, and does not "pollute" the mean field by altering brute-force. Finally, we note that modeling attackers and defenders separately would be warranted in a defensive setting where victims respond to the attack, where both group adapt their policies. However, VAI is an attack-only framework where victims are fixed and do not adapt. There is therefore no need to introduce a separate defender mean field, and the standard single mean-field formulation remains appropriate throughout.
>
>
> > Q2: Fixed Q-function assumption
>
> Please allow us to clarify that **there are two different value functions**. The first value function is the robust V function $V^i(s^i, \mu, \epsilon^i, \nu)$ **considering perturbations** gained from our regularized mean-field Bellman operator in Proposition 4.2. The second function is the Q function $Q^i(s^i, a^i, \mu, \nu)$ in Proposition 4.2, which is learned in a **cooperative** setting without perturbations. Since we already have a cooperative mean-field control policy as victim, we can learn this Q function via its policy rollout, which is done in the first K iterations in Algorithm 1. After Q converges, we query Q in the next K iteration to train robust V function.
>
> > Q3: optimality-preserving claimed in Prop. 4.5; approximation guarantees of VAI-Greedy and VAI-RL are not discussed.
>
> Our optimality-preserving claimed in Prop. 4.5 refers to the equivalence of the optimum solution, which means the ideal optimal solution of VAI is the optimal solution of HAD-MFC. However, VAI recovers the true optimum of HAD-MFC only when: (1) upper-level finds the optimal result of a NP-hard combinatorial problem, and (2) a perfect value function from the regularized Bellman operator. As such, our method must suffer from errors in approximating solution of NP-hard problems and function approximation error of neural networks and optimum is not guaranteed empirically.
>
> As for approximation guarantees of VAI-Greedy and VAI-RL, VAI-Greedy achieves an approximation ratio of 1-1/e on the upper-level due to submodularity. However, the approximation error of VAI-RL depends on learning a neural network, and its error follows the error of using RL to solve NP-hard problems, which is hard to bound.
>
> > Q4: runtime and computational scaling analysis in Taxi and Vicsek.
>
> As shown in the table below, the overall computation cost of our VAI is consistently on par with baselines in Taxi and Viscek. Due to the space limit, we will update the full table during the reviewer-author discussion time.
>
> It can be seen in the provided tables for the Taxi and Vicsek environments, the computational cost of our proposed VAI framework (both VAI-Greedy and VAI-RL) is consistently on par with existing baselines (e.g., Bi-Level, PIANO, RTCA). Notably, VAI exhibits a modest and stable increase in runtime. The overall complexity remains in the same order of magnitude as the baselines, confirming that VAI effectively maintains computational tractability while delivering its performance improvements in large-scale settings.
>
> > Q5: The paper just oscillates between positioning VAI as an attack method and a defense diagnostic tool. It is better to unify them to have a consistence presentation.
>
> Indeed, our VAI serves as an attack method and a diagnostic tool at the same time. Considering the nature of VAI, we will unify the expression as an attack in revision.
>
> > Q6: Metric plotted in Table 1 and statistical test.
>
> In Table 1, the metric is the reward given by the environment, and we follow their original settings. We conduct a paired-samples t-test and find the result is significant (p<.05) in 17 out of 18 tasks, except for one task where our method do not outperform baselines.
>
> > Q7: Too many typos.
>
> Thanks for pointing out. We will clear these typos in revision.

---

> > ### Author Rebuttal · Reviewer_ybT6 · 2026-04-03
> >
> > My comments are mostly addressed. Raised the score.

---

### Official Review · Reviewer_gUow · 2026-03-10

**Soundness:** 3
**Presentation:** 3
**Significance:** 3
**Originality:** 3
**Overall Recommendation:** 4
**Confidence:** 3

**Summary:**

This paper focuses on identifying vulnerable agents in large scale multi-agent systems. Concretely, it formulates the problem as a hierarchical adversarial decentralized mean field control problem, where the upper level’s task is to select the top-k vulnerable agents, and the lower level learns adversarial policies that cause the largest decrease in expected return. The problem is particularly challenging under large-scale conditions, because the upper level already involves an NP-hard combinatorial search  (i.e., selecting k agents from N),  and this difficulty is further increased by the need to jointly account for the coupled lower-level adversarial policy optimization. The authors propose to decouple this hierarchical process by modeling the upper level problem as an MDP with dense rewards, and reformulating it into a regularized mean-field Bellman operator, enabling a feasible manner to learn the most vulnerable agents without repeatedly training adversarial policies. Moreover, under certain assumptions, they prove that this decoupling strategy provably converges to the same global optima.

**Compliance With Llm Reviewing Policy:**

Affirmed.

**Final Justification:**

I found this paper to be a solid and technically sound contribution. The authors’ rebuttal sufficiently addressed my main concerns. In particular, the clarifications regarding the simulation setup and the experimental results resolved the uncertainties I had raised and reinforced my initial assessment. I'm maintaining my positive score.

**Key Questions For Authors:**

1.	In Table 1, the Taxi results show that, as the number of adversarial agents increases, the reported scores tend to increase rather than decrease across methods. This seems counterintuitive, since one might expect a larger number of adversarial agents to cause more severe performance degradation. Could you provide an explanation for this behavior?

2.	Could you provide some comparisons on the selected k agents for each algorithm? Do the selected agents differ substantially, or were the adversarial policies trained “better” for higher performing methods?

3.	I understand that the paper focuses on worst-case robustness, but I would be interested to hear your thoughts on whether the learned upper-level policy might generalize to suboptimal lower-level attackers as well. For instance, do you expect the vulnerable-agent ranking to remain similar when the lower-level adversarial policy is only partially optimized? This could be important in practical settings, where real-world attackers may not attain the true worst-case behavior.

**Limitations:**

Yes

**Strengths And Weaknesses:**

### Strengths
-	The authors present a technically sound method to identify vulnerable agents in large-scale multi-agent systems. Concretely, they reformulate the VAI problem in a hierarchical fashion, and decouple the upper and lower level problems by introducing a regularized mean-field Bellman operator which estimates the value function under different perturbation levels. This method allows efficiently solving the upper level NP-hard task, without needing to re-train adversarial policies for the selected k agents every time to evaluate the adversarial policies.
-	Experiment results demonstrate superior performance over existing VAI-related algorithms in Battle, Taxi Matching, and Vicsek environments.
-	The paper is presented in a clear manner, is well-organized and is easy to follow through the motivation behind the formulation of the problem, and their proposed solution.
-	To the best of my knowledge, the introduction of a regularized mean-field Bellman operator to estimate worst-case values for vulnerable-agent identification is completely novel.

### Weakness
-	Experiments are limited to simulated environments. Whether the proposed perturbed value function estimation works well in more realistic problems is unknown.
-	Controversial evaluation of VAI performance. From the results table, it is unclear whether the difference of negative rewards stems from correctly selecting the vulnerable agents, or successfully obtaining a more stronger adversarial policy.

### Miscellaneous
-	There seems to be a typo in the equation in Proposition 4.3. According to equation 10., it should be a “ – “ instead of a “+”. The resulting proofs remain the same, nonetheless.

---

> ### Author Rebuttal · Authors · 2026-03-29
>
> ### Official Response of Submission 14051 to Reviewer gUow
>
> We thank Reviewer gUow for appreciating the sound method, extensive experiment, clarity and novelty of our paper. A detailed response is provided below.
>
> ## Response
>
> > Q1: Experiments are limited to simulated environments. Whether the proposed perturbed value function estimation works well in more realistic problems is unknown.
>
> Thanks for mentioning this problem. Our Taxi environment mimics traffic management scenarios, which is connected to real-world settings. We are working on more seminal problems like social network and power nodes etc. and consider these results as our future work and leave this work to theoretical derivation.
>
>
> > Q2: It is unclear whether the difference of negative rewards stems from correctly selecting the vulnerable agents, or successfully obtaining a more stronger adversarial policy. Do the selected agents differ substantially, or were the adversarial policies trained “better” for higher performing methods?
>
> We have examined this and some prelinminary results are given in Section 5.2. As written in Line 397-404, right column, in Battle environment, our methods select agents at the right hand side which more frequently encounter enemies and deals more damage. The adversary also impact other benign agents at the right hand side so they failed to combat well when facing enemies. In Taxi, ride requests occur more frequently near the center, so adversaries mainly occur in the center, so other agents have to navigate to the center in order to get reward, but then gets blocked by the adversaries. In contrast, baseline methods like random and DC operates in a heuristic way, while learning based methods do not show identifiable patterns.
>
> As for the training process of adversarial policy, for each group of agents selected in our experiment, we train a worst-case adversarial policy based on these agents. As such, our method and baselines use the strongest lower-level policy, so the stronger result comes from the selected agents.
>
>
> > Q3: In Table 1, the Taxi results show that, as the number of adversarial agents increases, the reported scores tend to increase rather than decrease across methods. This seems counterintuitive, since one might expect a larger number of adversarial agents to cause more severe performance degradation. Could you provide an explanation for this behavior?
>
> Thanks for pointing this out. After checking it carefully we find it related to reward misspecification. There is a global multiplier in the environment about how uniform the agents are. With many cars being controlled by the adversary, the best policy for adversaries is not to do anything at the corner. However, in such case, the uniform multiplier term went up and the overall reward for the victims gets higher. We will correct the calculation of that multiplier in final revision.
>
> > Q4: I would be interested to hear your thoughts on whether the learned upper-level policy might generalize to suboptimal lower-level attackers as well.
>
> Thanks for this great question. In the agent-wise perspective, in Proposition 4.4, we connect our theory to worst-case robustness within perturbations in a certain range. As such, we hypothesize that agents with more worst-case vulnerability should be more sentitive to noise as well. In the structure-wise perspective, if a certain structure is vulnerable to worst-case attack, then we hypothesize that these agents forming the structure should also be vulnerable to attack.
>
> However, from the theoretical perspective, our analysis is based on worst-case attack and we cannot guarantee exactly the identified vulnerable agents also generalize to lower-level perturbations that is not the worst-case.
>
> > Q5: Typo in the equation in Proposition 4.3.
>
> Thanks for pointing out. We will correct this in final revision.

---

> > ### Author Rebuttal · Reviewer_gUow · 2026-04-03
> >
> > Thank you for the clarification. My concerns are sufficiently addressed and I will maintain my rating.

---

### Official Review · Reviewer_34Pv · 2026-03-10

**Soundness:** 2
**Presentation:** 2
**Significance:** 3
**Originality:** 3
**Overall Recommendation:** 4
**Confidence:** 3

**Summary:**

This paper studies vulnerable agent identification in large-scale mean-field MARL: selecting a subset of agents whose compromise degrades system performance most. The paper formulates the problem as a hierarchical adversarial decentralized mean-field control problem, with an upper level that chooses attacked agents and a lower level that learns their worst-case adversarial policy. The key idea is to avoid solving the lower-level attack for every candidate subset by deriving a regularized Bellman-style value estimator from benign trajectories, then using that estimator to define dense rewards for a sequential upper-level selection MDP. The paper proposes two selection algorithms, VAI-Greedy and VAI-RL, and evaluates them on Battle, Taxi, and Vicsek. Empirically, the method is strong and often substantially outperforms adapted baselines.

**Compliance With Llm Reviewing Policy:**

Affirmed.

**Final Justification:**

The rebuttal addressed my main concerns, so I raised my score.

**Key Questions For Authors:**

The questions are the same as the weaknesses discussed above.

**Limitations:**

Partially. The paper includes an impact statement and briefly notes that the method can help defenders but could also help attackers. However, authors are encouraged to add some discussion on the limitations of the proposed method.

**Strengths And Weaknesses:**

Strengths:
1. Robustness and security questions in large-scale MARL are underexplored, and identifying which agents are most critical under worst-case compromise is of interest.
2. Combining mean-field adversarial control with a sequential vulnerable-agent selector is a creative way to address the otherwise intractable bi-level problem. The decomposition into lower-level value estimation plus upper-level selection is intuitive and, at a high level, appealing.
3. The paper evaluates on two mean-field MARL environments plus one rule-based system, and the improvements in several settings are large.

Weaknesses:
1. My main concern is that the paper's strongest theoretical claim appears overstated. The manuscript repeatedly mentions the lower-level transform and the overall decomposition as exact (Remark 3; Proposition 4.5). However, Appendix A. 4 explicitly introduces a "fixed Q-function assumption": the derivation assumes access to a fixed $Q(s, a, \mu, \nu)$ from policy evaluation and treats the reward-plustransition term as constant with respect to perturbations during the robust policy improvement derivation. That is a major modeling simplification. It seems to justify an exact transform of a surrogate problem with frozen $Q$, not obviously the original lower-level adversarial control problem in which perturbations affect transitions and mean-field quantities. As written, Proposition 4.2 and Proposition 4.5 do not seem to support the optimality claim.
2. Since the paper claims the decomposition preserves the true optimum, I would have expected at least one small-scale exhaustive validation: enumerate all vulnerable-agent subsets in a toy setting, compare to the exact bi-level optimum, and measure whether the proposed selector and lower-level value proxy recover the same solution. Without that, the central theorem is not empirically stress-tested.
3. Baseline fairness. In Appendix C.2, the paper states that for PIANO, the upper-level reward is approximated using lower-level attackers executing a random policy because the original setting does not include adversarial policy learning. That adaptation may significantly handicap the baseline. Likewise, graph-based baselines are forced to operate on a heuristic observation graph in environments that do not naturally provide such a graph. The paper should be more careful in framing these comparisons.
4. The presentation needs improvement. Sections 4.1–4.2 and Appendix A.4 are difficult to follow, notation is dense, and there are several writing issues and inconsistencies. The paper has a strong idea, but the current exposition makes it hard to assess which claims are exact, which are approximate, and which are empirical heuristics.

---

> ### Author Rebuttal · Authors · 2026-03-29
>
> ### Official Response of Submission 14051 to Reviewer 34Pv
>
> We thank Reviewer 34Pv for appreciating the novelty, appealing technical method and large-scale experiment of our paper. A detailed response is provided below.
>
> > Q1: Theoretical claim appears overstated. Manuscript mentioned exact transform but Appendix mentioned "fixed Q-function assumption".
>
> Please allow us to clarify that **there are two different value functions**. The first value function is the robust V function $V^i(s^i, \mu, \epsilon^i, \nu)$ **considering perturbations** gained from our regularized mean-field Bellman operator in Proposition 4.2. The second function is the Q function $Q^i(s^i, a^i, \mu, \nu)$ in Proposition 4.2, which is learned in a **cooperative** setting without perturbations, and is a consequence of our theoretical derivation. Since we already have a cooperative mean-field control policy as victim, we can learn this Q function via its policy rollout. Afterwards, we query from ||Q||_q in case of need.
>
> > Q2: Since the paper claims the decomposition preserves the true optimum, I would have expected at least one small-scale exhaustive validation: enumerate all vulnerable-agent subsets in a toy setting, compare to the exact bi-level optimum, and measure whether the proposed selector and lower-level value proxy recover the same solution. Without that, the central theorem is not empirically stress-tested.
>
> We provide an analytical verification on a tractable #-size infinite-grid setting. Consider a power grid where each node is an agent with state {0, 1} and action {on, off}. All nodes begin in state 0 and action on (reward +1). Choosing off incurs a penalty equal to itself plus the number of neighbours also choosing off; state 1 doubles all rewards, marking a critical node. With 5 attackers and one critical node (state 1), the exact optimum is to select that critical node and its 4 neighbours all taking off, with critical node having reward -5 (itself and neighgbours going off) * 2 (state=1) = -10.
>
> Now consider our algorithm with $\ell_infty$ norm for simplicity. When $\epsilon^i = 1$ (node going off) and $\xi = 0$ (all neighbours still working), the robust V function with state=1 receives worst-case as -2, while others with state=1 receives worst-case as -1, so VAI-RL should choose node with state=1. Afterwards, after selecting a node of its neighbour, the worst-case adds by -1 (itself going off) - 2 (near state=1), so VAI-RL should choose the neighbour nodes. Finally, the node with state=1 and its neighbours should have been chosen.
>
> Note that this is a highly simplified scenario. As stated in Proposition 4.5, our algorithm recovers the true optimum of HAD-MFC only when: (1) upper-level finds the optimal result of a NP-hard combinatorial problem, and (2) a perfect value function from the regularized Bellman operator. As such, our method must suffer from errors in approximating solution of NP-hard problems and function approximation error of neural networks and optimum is not guaranteed empirically.
>
> > Q3: Baseline fairness. In Appendix C.2, the paper states that for PIANO, the upper-level reward is approximated using lower-level attackers executing a random policy because the original setting does not include adversarial policy learning. That adaptation may significantly handicap the baseline. Likewise, graph-based baselines are forced to operate on a heuristic observation graph in environments that do not naturally provide such a graph. The paper should be more careful in framing these comparisons.
>
> In the original **PIANO** setting, the upper-level reward is provided immediately by the environment after agent selection. In our setting, obtaining the true attack reward requires training a lower-level adversarial policy, which takes several hours per candidate group, making exhaustive upper-level search with full lower-level evaluation computationally infeasible. This is the core difficulty our Proposition 4.2 addresses, by replacing the expensive lower-level training with a closed-form value proxy. For PIANO specifically, we use a random policy as the upper-level reward proxy during selection; after the final group is selected, we train a full adversarial policy on that group to obtain the reported attack result. This is the fairest adaptation possible given PIANO's original design.
>
> As for **Graph-based baselines**, our mean-field environments do not provide an explicit graph structure. Rather than omitting these baselines or the graph structure entirely, we construct a heuristic graph from observation ranges, giving these methods the best available structural information.
>
> > Q4: The presentation needs improvement. The paper has a strong idea, but the current exposition makes it hard to assess which claims are exact, which are approximate, and which are empirical heuristics.
>
> Many thanks for appreciating our idea. We will update the writing thoughout our paper in the final version.

---

> > ### Author Rebuttal · Reviewer_34Pv · 2026-03-31
> >
> > Thanks for the detailed response. I have increased my score.

---

### Decision · Program_Chairs · 2026-04-30

**Decision:**

Accept (regular)

**Comment:**

This paper makes a novel and meaningful contribution to vulnerable-agent identification in large-scale mean-field MARL through a regularized mean-field Bellman operator derived via Fenchel–Rockafellar duality. The rebuttal satisfactorily clarified the main concerns regarding the fixed-Q assumption and the scope of the optimality claim, making clear that the theory applies to an idealized setting while the practical method is approximate. Although the presentation could be improved, the technical idea is strong, the empirical results are convincing, and I recommend acceptance.